# PLK-1 promotes the merger of the parental genome into a single nucleus by triggering lamina disassembly

Griselda Velez-Aguilera[1], Sylvia Nkombo Nkoula[1], Batool Ossareh-Nazari[1], Jana Link[2], Dimitra Paouneskou[2], Lucie Van Hove[1], Nicolas Joly[1], Nicolas Tavernier[1], Jean-Marc Verbavatz[3], Verena Jantsch[2], Lionel Pintard[1]*

[1]Programme Equipe Labéllisée Ligue Contre le Cancer - Team Cell Cycle & Development - Université de Paris, CNRS, Institut Jacques Monod, Paris, France; [2]Department of Chromosome Biology, Max Perutz Laboratories, University of Vienna, Vienna Biocenter, Vienna, Austria; [3]Université de Paris, CNRS, Institut Jacques Monod, Paris, France

**Abstract** Life of sexually reproducing organisms starts with the fusion of the haploid egg and sperm gametes to form the genome of a new diploid organism. Using the newly fertilized *Caenorhabditis elegans* zygote, we show that the mitotic Polo-like kinase PLK-1 phosphorylates the lamin LMN-1 to promote timely lamina disassembly and subsequent merging of the parental genomes into a single nucleus after mitosis. Expression of non-phosphorylatable versions of LMN-1, which affect lamina depolymerization during mitosis, is sufficient to prevent the mixing of the parental chromosomes into a single nucleus in daughter cells. Finally, we recapitulate lamina depolymerization by PLK-1 in vitro demonstrating that LMN-1 is a direct PLK-1 target. Our findings indicate that the timely removal of lamin is essential for the merging of parental chromosomes at the beginning of life in *C. elegans* and possibly also in humans, where a defect in this process might be fatal for embryo development.

*For correspondence:
Lionel.PINTARD@ijm.fr

Competing interests: The authors declare that no competing interests exist.

## Introduction

After fertilization, the haploid gametes of the egg and sperm have to come together to form the genome of a new diploid organism. In the zygotes of humans, but also in *Caenorhabditis elegans*, parental chromosomes are initially in separate pronuclei each surrounded by a nuclear envelope (NE) and meet for the first time during the first mitotic division. For this mitosis to occur properly, the NE of the pronuclei must be disassembled in a timely manner (*Rahman et al., 2020*). This step is essential to permit the mingling of the maternal and paternal chromosomes and their capture by the microtubules of the developing spindle.

The newly fertilized one-cell *C. elegans* embryo provides a dynamic developmental context for the investigation of the mechanism regulating nuclear envelope breakdown (NEBD) (*Figure 1—figure supplement 1A*) and for review (*Oegema, 2006*; *Cohen-Fix and Askjaer, 2017*; *Pintard and Bowerman, 2019*). After fertilization, the egg and sperm pronuclei meet near the posterior pole of the embryo. After rotation and centration of the nucleo-centrosomal complex, the NE breaks down in the vicinity of the centrosomes and between the juxtaposed pronuclei. This allows the capture of the chromosomes by the microtubules and the mixing of the parental chromosomes (*Figure 1—figure supplement 1A*). The breakdown of the envelope located between the two pronuclei starts with the formation of a membrane gap (*Audhya et al., 2007*), most likely triggered by egg and sperm chromosomes that come into contact with the NE. Accordingly, gap formation depends on proper chromosome alignment at the metaphase plate (*Rahman et al., 2015*). Complete disassembly of the

NE occurs later at the metaphase-to-anaphase transition in *C. elegans* embryos (*Lee et al., 2000*). NEBD is thus spatially regulated in the early embryo but the underlying mechanisms are still unclear.

If the NE between the two pronuclei persists through anaphase, as is the case in mutant embryos with delays or defects in NEBD, the two sets of parental chromosomes remain physically separated during the first mitosis. Consequently, the maternal and paternal chromosomes segregate into two separate DNA masses at each pole of the spindle. This in turn leads to the formation of two separate nuclei in each cell of the two cell embryo, called the 'paired nuclei' phenotype (*Figure 1—figure supplement 1A*; *Rahman et al., 2015*). Several studies have documented conditions that affect NEBD and give rise to the paired nuclei phenotype (*Audhya et al., 2007*; *Golden et al., 2009*; *Gorjánácz and Mattaj, 2009*; *Bahmanyar et al., 2014*; *Galy et al., 2008*; *Rahman et al., 2015*; *Martino et al., 2017*). In particular, this phenotype is typically associated with inactivation of the mitotic Polo-like kinase (PLK-1 in *C. elegans*, Plk1 in humans and Polo in *D. melanogaster*) (*Rahman et al., 2015*; *Martino et al., 2017*), which regulates many aspects of mitotic entry and progression (*Archambault and Glover, 2009*; *Zitouni et al., 2014*). However, the critical targets of PLK-1 involved in this process are largely unknown. Identifying these targets is of primary importance to decipher how PLK-1 promotes the merger of the maternal and paternal genomes into a single nucleus after fertilization.

Consistent with a role in NEBD, PLK-1, is specifically recruited to the NE in prophase just before NEBD via its C-terminal Polo-Box domain (PBD), both in human cells (*Linder et al., 2017*) and in *C. elegans* embryos (*Martino et al., 2017*). The PBD domain binds to specific phosphorylated sequence motifs (polo-docking sites) that are created by other priming kinases (non-self-priming) or are self-primed by PLK-1 itself, thus providing an efficient mechanism to regulate PLK-1 subcellular localization and substrate selectivity in space and time (*Park et al., 2010*). At the NE, PLK-1 could potentially regulate all the steps of NEBD, in particular, depolymerization of the nuclear lamina.

The lamina is a rigid proteinaceous meshwork underlying the inner nuclear membrane, bridging the nuclear envelope and chromatin (*Aaronson and Blobel, 1975*; *Goldman et al., 1986*; *Aebi et al., 1986*; *Burke and Stewart, 2013*). All lamins are type V intermediate filaments composed of an unstructured N-terminal head, a central α-helical coiled-coil rod domain, and a C-terminal tail domain (*Stuurman et al., 1998*). Lamin filaments are composed of lamin dimers, which assemble into dimeric head-to-tail polymers. Two head-to-tail polymers then assemble laterally into a protofilament with a uniform shape rod of around 3.5 nm in diameter (*Turgay et al., 2017*; *Ahn et al., 2019*). Lamin filaments depolymerization is triggered by mitotic phosphorylation of the head and tail domains (*Gerace and Blobel, 1980*; *Heald and McKeon, 1990*; *Mühlhäusser and Kutay, 2007*) but the regulation of lamin phosphorylation is complex and not fully understood (*Machowska et al., 2015*).

Conserved Cdk (Cyclin-dependent kinase) consensus motifs (S/T-P) are present in most lamins and their phosphorylation is important for lamina disassembly in vitro and in vivo (*Ward and Kirschner, 1990*; *Heald and McKeon, 1990*; *Peter et al., 1990*; *Peter et al., 1991*; *Dessev et al., 1991*; *Mehsen et al., 2018*). However, in most studies conducted in vivo, the impact of lamin-phosphosite mutations on lamina disassembly during mitosis has been investigated only in the presence of endogenous phosphorylatable lamin. In this context, the resulting phenotype is highly dependent on the level of overexpression of the mutated lamin over the wild-type version, which makes the precise analysis of the contribution of the phosphosites to lamina disassembly particularly challenging. Based on a similar overexpression approach, Protein Kinase C (PKC) has been proposed to act in concert with Cdk1 to promote phosphorylation-dependent lamin B1 disassembly through lipid signaling in human cells (*Mall et al., 2012*). These observations suggest that multiple kinases cooperate to promote efficient mitotic lamin disassembly.

Although Plk1 has not been directly implicated in lamin phosphorylation during mitosis, the lamina persists during mitosis upon *plk-1* inactivation, resulting in a paired nuclei phenotype, consistent with a possible role of PLK-1 in lamina depolymerization in *C. elegans*. Moreover, partial inactivation of *lmn-1*, encoding the unique lamin isoform (B-type) expressed in *C elegans* (*Liu et al., 2000*), suppresses the paired nuclei phenotype of *plk-1* temperature-sensitive (ts) mutant embryos (*Rahman et al., 2015*; *Martino et al., 2017*).

Here we show that PLK-1 directly phosphorylates LMN-1 at multiple sites in the head and tail domains to trigger lamina depolymerization. Accordingly, the expression of non-phosphorylatable versions of LMN-1 prevents lamina depolymerization during mitosis, which is sufficient to induce the

formation of 2cell embryos with a paired nuclei phenotype. Furthermore, we reconstitute the depolymerization of pre-assembled LMN-1 filaments by PLK-1 in vitro. Our findings demonstrate that PLK-1 is a direct lamin kinase *in C. elegans* and possibly also in other organisms.

## Results

### PLK-1 phosphorylates the *C. elegans* lamin LMN-1 at multiple sites in vitro

To determine if LMN-1 is a direct PLK-1 substrate in *C. elegans*, we tested whether PLK-1 directly phosphorylates LMN-1 in vitro. As lamina disassembly is regulated by phosphorylation of the head and the tail domains, we specifically concentrated on these two regions of LMN-1. We produced and purified the LMN-1 head (LMN-1[H] aa 1–47) and tail domains (LMN-1[T] aa 386–566) fused to the Glutathion S-transferase (GST) in *E. coli* and performed in vitro kinase assay using *Ce* PLK-1 kinase (*Figure 1A and B*), purified from insect Sf9 cells (*Tavernier et al., 2015*). We then separated the product of the reaction on Phos-Tag SDS-PAGE to resolve the phosphorylated forms of GST-LMN-1 fragments. As shown in *Figure 1C*, PLK-1-mediated phosphorylation induced a mobility shift of the GST-LMN-1 [H] and [T] fragments with the appearance of multiple bands on Phos-Tag SDS-PAGE, but not GST. In a similar assay, the other mitotic kinases Aurora A and Cyclin B-Cdk1 failed to induce a mobility shift of GST-LMN-1 [H] (*Figure 1—figure supplement 1B*) indicating that PLK-1, but not these other mitotic kinases, phosphorylates LMN-1 in vitro, possibly at multiple residues.

We then subjected the LMN-1 head and tail fragments, phosphorylated in vitro by PLK-1, to mass spectrometry analysis and identified at least 15 (S/T) phosphorylated residues distributed in the head (S18, S21, S22, S24, S32, S35, T36, T40 and S41) and the tail domains just upstream the Ig-Fold domain (T390, T397, S398, S403, S405, S406) (*Figure 1D and E* and *Supplementary file 1*). Sequence analysis revealed that most of these sites match the consensus motif for Plk1 phosphorylation [L(Φ)-D/E/N/Q-X-pS/pT-L(Φ)] or [pS/pT-F] (*Supplementary file 2*; *Santamaria et al., 2011*; *Kettenbach et al., 2011*).

To confirm that PLK-1 phosphorylates LMN-1 on these sites, we substituted serines and threonines with non-phosphorylatable alanines and repeated the in vitro kinase assays in the presence of radiolabeled ATP [$\gamma^{32}$P] (*Figure 1F and G*). As substrates, we used the same LMN-1 fragment for the head ([H] aa 1–47), but a version of the tail excluding the Ig-Fold domain (LMN-1 [T1] aa 386–437). Different mutated versions of these LMN-1 [H] and [T1] fragments, with different serine and threonine substitutions, displayed a significantly dampened phosphorylation by PLK-1, indicating that these residues are indeed the major PLK-1 phosphorylation sites on the head and the tail domains of LMN-1 (*Figure 1F and G*). In particular, substitution of the six serines and threonines in the tail fragment entirely prevented phosphorylation by PLK-1 (*Figure 1I*, *Figure 1—source data 1* - Table 1). Substitution of seven of the nine sites in the head domain dampened but did not fully abolish phosphorylation of LMN-1 [H] 7A (*Figure 1H*, *Figure 1—source data 1*), which is still phosphorylated on T40 and S41 residues, as suggested by the mass spectrometry results (*Supplementary file 1*). Consistent with T40 and S41 being phosphorylated by PLK-1 in the LMN-1 [H] 7A fragment, this fragment was no longer retarded on SDS-PAGE after substitution of these two residues by alanines (*Figure 1J*). Taken together, these studies indicate that PLK-1 directly phosphorylates LMN-1 head and tail domains at multiple sites in vitro.

### PLK-1 PBD interacts with lamins from several species via self and non-self-priming and binding mechanisms

Phosphorylation of lamin head and tail domains by Cyclin B-Cdk1 kinase has been shown previously to contribute to lamina disassembly during mitosis in human cells (*Heald and McKeon, 1990*) and more recently in *Drosophila* S2 cells (*Mehsen et al., 2018*). Cyclin-Cdk typically phosphorylates substrates on the minimal consensus Ser/Thr-Pro (S/T-P) motif (*Nigg, 1993*). Multiple protein sequence alignments between vertebrates, *Drosophila* and nematode lamins, revealed that *C. elegans* LMN-1, and more broadly lamins in nematodes, do not harbor any Cdk S/T-P motifs in the head or in the tail domains (*Figure 2A*). Proline residues, highlighted in red in the protein sequence alignment presented in *Figure 2A*, are strikingly absent in the head and tail domains of lamins from nematodes, consistent with the observation that Cyclin B-Cdk1 failed to phosphorylate GST-LMN-1[H]

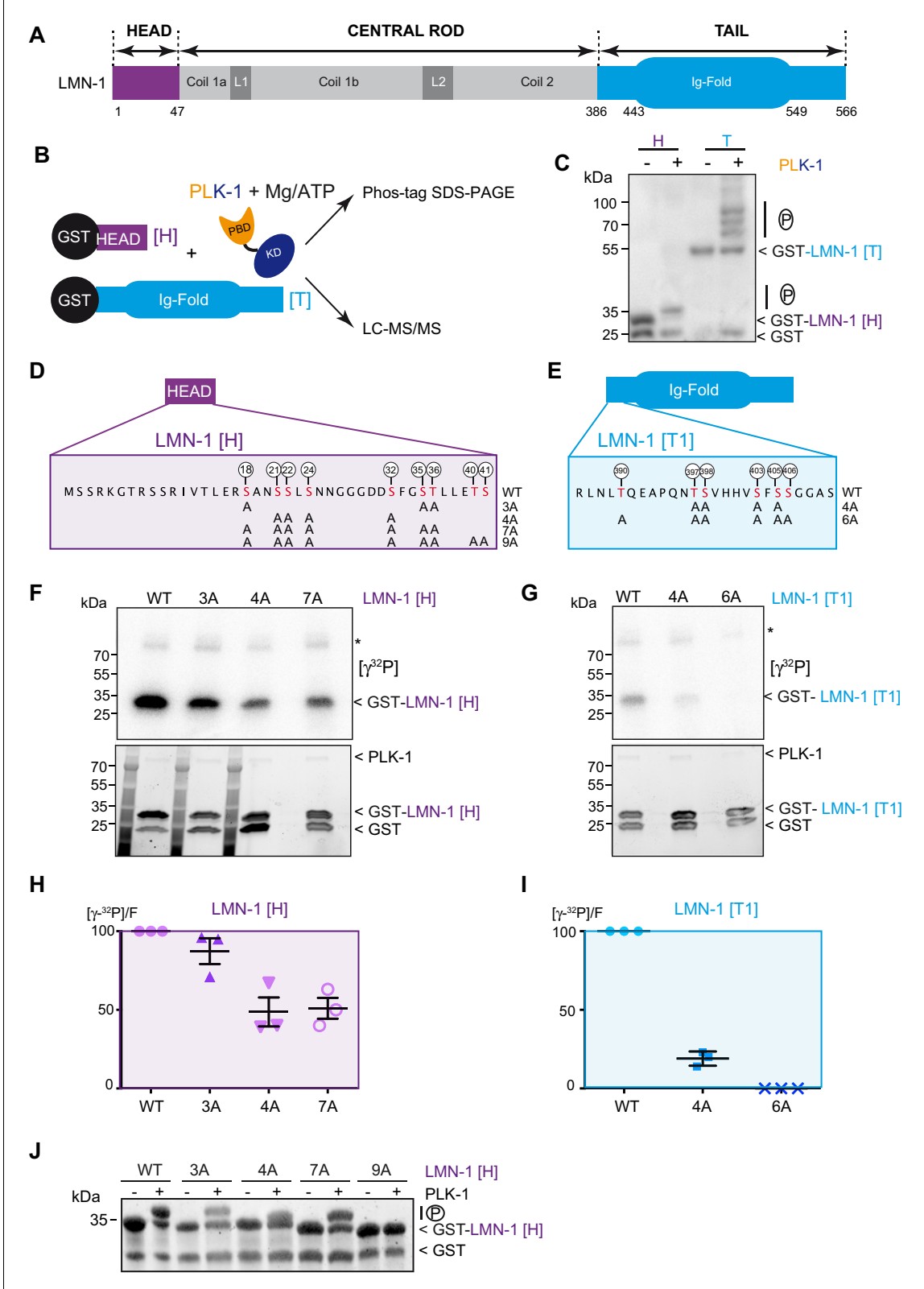

**Figure 1.** PLK-1 phosphorylates LMN-1 head and tail domains at multiple sites in vitro. (**A**) Schematic of LMN-1 domain structure. Similar to lamins from other species, LMN-1 has a tripartite structure consisting of non-α-helical N-terminal head (violet) and C-terminal tail (blue) domains flanking an α-helical central rod domain. The C-terminal tail contains the Ig-fold motif. (**B**) Schematic of the approach used to investigate whether PLK-1 directly phosphorylates LMN-1 head and tail domains and to map the phosphorylated sites by tandem mass spectrometry (LC-MS/MS). (**C**) In vitro kinase assay

*Figure 1 continued on next page*

*Figure 1 continued*

was performed with *C. e* PLK-1 and the GST-LMN-1 head or tail as substrates. The samples were subjected to Phos-Tag SDS-PAGE followed by a western blot analysis using GST antibody. (**D–E**) Protein sequence of the head and tail domains with the residues phosphorylated by PLK-1 in red. The position of the phosphorylated residues is also indicated by white circles. The different non-phosphorylatable versions of LMN-1[H] 3A, 4A, 7A, 9A and LMN-1[T1] 4A, 6A with the positions of the alanine substitutions are presented below the protein sequence. (**F–G**) In vitro kinase assays using PLK-1 and LMN-1 WT or mutated fragments (LMN-1[H] and [T1]) as substrates. An autoradiograph of the SDS-PAGE showing γ-[$^{32}$P] incorporation in LMN-1 (upper panel). Coomassie brilliant blue (CBB) staining of the same SDS-PAGE (bottom panel). Asterisk marks autophosphorylated PLK-1. GST is present as an impurity from the production of GST-LMN-1 fragments. (**H–I**) The graphs correspond to the quantification of the radioactivity incorporated into LMN-1 [H] and [T1] fragments divided by the total amount of LMN-1 fragments quantified using tryptophan fluorescence (Stain-Free; Bio-Rad). The ratio obtained for the WT fragments was arbitrary defined as 100. The quantification of three independent experiments is presented. Error bars represent the standard deviation. (**J**) In vitro kinase assay was performed with *C. e* PLK-1 and the GST-LMN-1 [H] WT or mutated fragments as substrates. The samples were subjected to SDS-PAGE and the gel was revealed using tryptophan fluorescence (Stain-Free; Bio-Rad).

The online version of this article includes the following source data and figure supplement(s) for figure 1:

**Source data 1.** related to *Figure 1*: Quantification of the in vitro kinase assays.
**Figure supplement 1.** PLK-1 but not Aurora A or Cyclin B/Cdk1 phosphorylates LMN-1[H] in vitro.

(*Figure 1—figure supplement 1B*). Instead, several sites phosphorylated by PLK-1 on *C. elegans* LMN-1 are located in equivalent position to SP or TP sites phosphorylated by Cyclin-Cdk1 kinase on lamins from other species (S18, S35, T36, T390 *Figure 2A*, black arrows at bottom of alignment) suggesting that PLK-1 might be the main kinase in *C. elegans* phosphorylating LMN-1 to trigger lamina disassembly during mitosis and to promote pronuclear fusion.

Protein sequence analysis also revealed that PLK-1 phosphorylates LMN-1 on putative polo-docking sites (*Figure 2A*), which match the consensus for self-priming sites, in the head (20-SSL-24, 34-STL-38) and in the tail domain (404-SSG-408) suggesting that PLK-1 might recruit itself to the lamin via a self-priming mechanism. We tested this hypothesis using a Far-Western ligand-binding assay. We pre-phosphorylated the head or the tail domain of LMN-1 using PLK-1 and then tested their ability to interact with the PLK-1 PBD. As shown in the *Figure 2—figure supplement 1A* (panels a, b), the LMN-1 head, but not the tail domain, pre-phosphorylated by PLK-1 readily interacted with the PLK-1-PBD (panel a), but not with a version of the PBD carrying mutated phospho-pincers (panel b), demonstrating that this interaction is phospho-dependent. To test if the two polo-docking sites defined by sequence analysis accounted for PBD binding, we analyzed different LMN-1 [H] fragments (3A, 4A, 7A) containing alanine substitutions in the predicted polo-docking sites (*Figure 2B*, panels a, b and *Figure 2—figure supplement 1B*). The introduced alanine substitution in one or the other polo-docking site did not prevent binding (*Figure 2B*, panel b, lanes 6 and 8) but alanine substitution of both sites abrogated binding of LMN-1 to the PLK-1 PBD (lane 10). We conclude that PLK-1 phosphorylates LMN-1 on two polo-docking sites in the head domain for self-priming and binding in vitro.

Protein sequence analysis suggested that Plk1/Polo might also interact with lamins from other species, as vertebrate and *Drosophila* lamins contain putative polo-docking sites in the head domain (*Figure 2A*). However, instead of matching the consensus for PLK-1 self-priming phosphorylation and binding (S-pS-X), these sites match the consensus for non-self-priming and binding (S-pT/pS-P). Notably, 12 out of 21 lamins from vertebrates and *Drosophila* harbor the canonical STP site in the head domain (*Figure 2A*), which perfectly matches the consensus for non-self-priming and binding (*Park et al., 2010*). *Drosophila* lamin Dm0 contains two of these sites while human lamin A/C contains one (*Figure 2C and D* panel a). As these polo-docking sites are modified by Cyclin B-Cdk1 during mitosis, both in *Drosophila* (*Machowska et al., 2015*; *Mehsen et al., 2018*) and in human cells (*Dephoure et al., 2008*; *Chen et al., 2013*), we reasoned that their phosphorylation might promote Plk1 binding via a non-self-priming mechanism. We tested this hypothesis using the Far-western ligand-binding assay with *Drosophila* and human lamin A/C head fragments (*Figure 2C and D* respectively and *Figure 2—figure supplement 1C and D*). As shown in the *Figure 2C and D*, a robust interaction between human or *Drosophila* lamin phosphorylated by Cyclin B-Cdk1 and the Plk1 PBD was observed by the Far-Western ligand-binding assay (*Figure 2C and D*, panel b lane 2). This interaction is phospho-dependent because no binding was observed using a mutated version of the PBD that is unable to recognize phosphopeptides (*Figure 2—figure supplement 1C and D*, panels b). The canonical non-self-priming STP and SSP polo-docking sites accounted for all the PBD

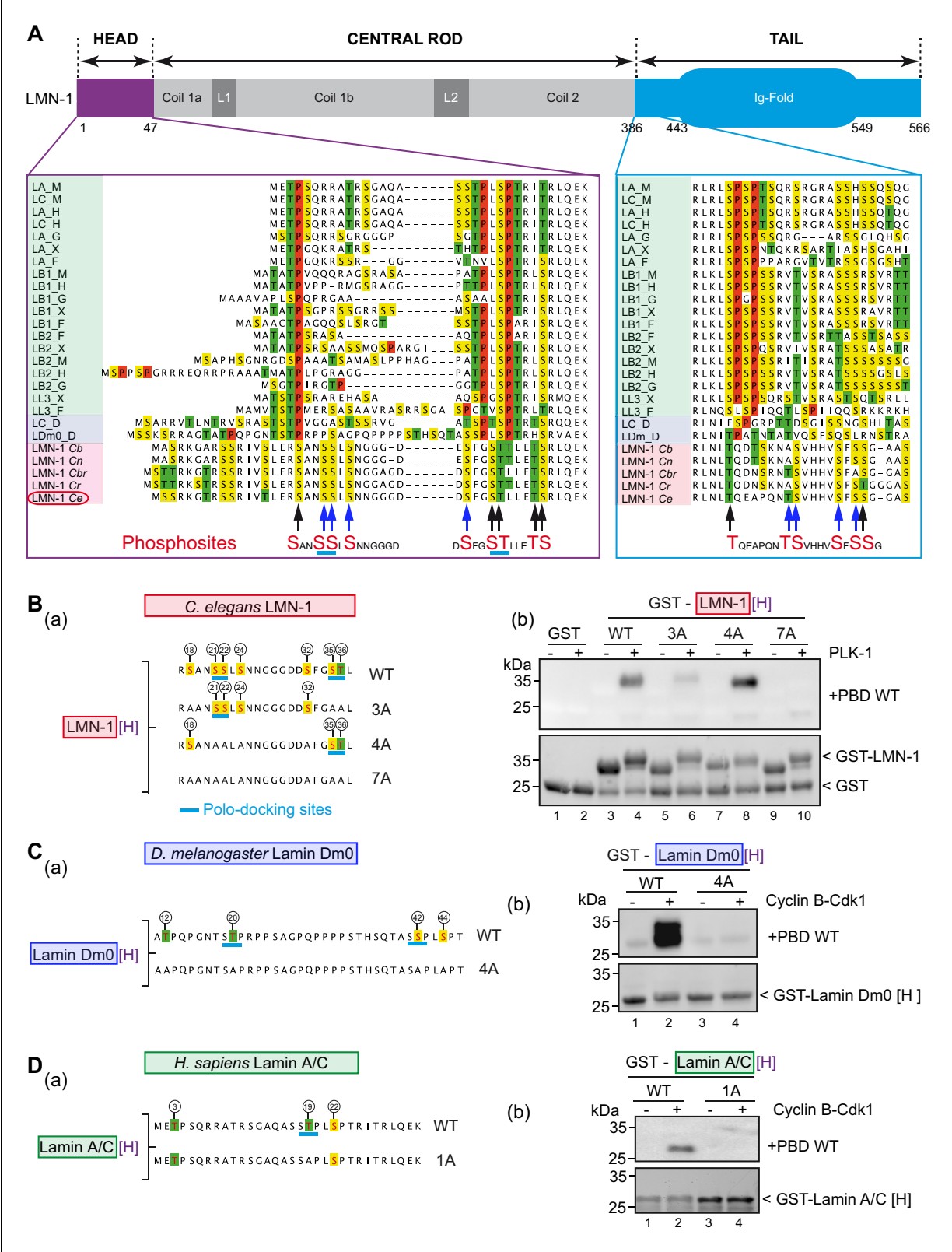

**Figure 2.** PLK-1 interacts with lamins from several species via self and non-self-priming and binding mechanisms. (**A**) Schematics of LMN-1 and multiple protein alignments of the head (violet) and the tail (blue) domains of Ce LMN-1 (surrounded in red, bottom lane) with lamins from vertebrates (light green), *Drosophila melanogaster* (light blue) and other nematodes (light red). Serine residues are highlighted in yellow, Threonine in green and Proline in red. The sequence correspondence and accession numbers are provided in the **Supplementary file 3**. The black and blue arrows point the C. e

*Figure 2 continued on next page*

*Figure 2 continued*

LMN-1 residues phosphorylated by PLK-1 in red at the bottom of the alignment. The blue arrows point residues previously shown to be phosphorylated in the *C. elegans* germline. Putative polo-docking sites are underlined in light blue. (B) (a) Schematics and sequence of the GST-LMN-1 [H] fragment WT or with the indicated substitution of serine and threonine residues by non-phosphorylatable alanine (3A, 4A, and 7A). Polo-docking sites are underlined in light blue. (b) In vitro kinase assay was performed with PLK-1 and the GST-LMN-1 [H] fragments as substrate. The samples were subjected to SDS-PAGE, followed by a Far-Western ligand-binding assay using the Polo-box domain fused to GST (+PBD, upper panel). The bottom panel shows the Stain-Free Blot (Chemidoc, Bio-Rad) of the same membrane. The full experiment is presented in *Figure 2—figure supplement 1B*. (C) (a) Schematics and sequence of *Drosophila melanogaster* lamin Dm0 head domain WT or 4A fused to GST. The sites phosphorylated by Cyclin-Cdk1 are circled (*Mehsen et al., 2018*). These four sites are substituted by alanine in the 4A construct. The canonical polo-docking sites are underlined in light blue. (b) In vitro kinase assay was performed with Cyclin-Cdk1 and the GST-lamin Dm0 [H] fragments as substrate. The samples were subjected to SDS-PAGE, followed by a Far-Western ligand-binding assay using the Polo-box domain fused to GST (+GST-PBD, upper panel). The bottom panel shows the Stain-Free Blot (Chemidoc, Bio-Rad) of the same membrane. The full experiment is presented in *Figure 2—figure supplement 1C*. (D) (a) Schematics and sequence of human lamin A/C head domain. This fragment contains three S/T-P sites phosphorylatable by Cyclin-Cdk1 kinase including one polo-docking site (underlined in light blue). This specific site is substituted by alanine in the 1A version. (b) In vitro kinase assay was performed with cyclin-cdk1 and the GST-lamin A/C [H] WT or 1A fragments as substrate. The samples were subjected to SDS-PAGE, followed by a Far-Western ligand-binding assay using the Polo-box domain fused to GST (+GST-PBD, upper panel). The bottom panel shows the Stain-Free Blot (Chemidoc, Bio-Rad) of the same membrane. The full experiment is presented in *Figure 2—figure supplement 1D*.

The online version of this article includes the following figure supplement(s) for figure 2:

**Figure supplement 1.** Lamins from several species interact with the Plk1 PBD in a phospho-dependent manner.

binding activity as substitution of the threonine and serine to alanine abrogated binding under these conditions (*Figure 2C and D*, panel b lane 4). Taken together, these results indicate that Plk1 binds pre-phosphorylated head domains of lamins from several species via self and non-self-priming mechanisms.

## PLK-1 phosphorylates LMN-1 to promote lamina depolymerization during mitosis in early embryos

We next investigated whether PLK-1, once recruited to the lamina, phosphorylates LMN-1 to promote lamina disassembly during mitosis. Notably, among the fifteen LMN-1 residues identified as being phosphorylated by PLK-1 in vitro, eight have been previously shown to be phosphorylated in vivo in the *C. elegans* germline (S21, S22, S24, S32, S397, S398, S403, S405 (*Supplementary file 2*) and sites indicated by blue arrows in *Figure 2A*; *Link et al., 2018*). These sites are presumably also phosphorylated in early embryos. Accordingly, indirect immunofluorescence using a phosphospecific antibody directed against the phosphorylated serine 32 of LMN-1 revealed that this site is readily modified in early embryos in a PLK-1-dependent manner (*Figure 3—figure supplement 1*).

In the germline, LMN-1 phosphorylation by the master meiotic kinases CHK-2, and to a lesser extent PLK-2, contribute to transient lamina disassembly, facilitating homologous chromosome movement during meiotic prophase (*Link et al., 2018*) and for review (*Zetka et al., 2020*). Phosphorylation of these eight sites also impacts on lamina depolymerization during mitosis. Indeed, lamina disassembly is delayed in embryos expressing a *gfp::lmn-1* transgene encoding a LMN-1 version carrying these eight serine residues replaced by alanines ([LMN-1 8A], *Figure 3*) in a *lmn-1* null deletion strain (*Link et al., 2018*). However, as eight of the fifteen LMN-1 sites phosphorylated by PLK-1 in vitro have been mutated to alanine on the *gfp::lmn-1* 8A transgene, we reasoned that *plk-1* inactivation in *gfp::lmn-1* 8A mutant embryos should further stabilize the lamina network during mitosis. To test this hypothesis, we used spinning disk confocal microscopy to monitor and quantify lamina disassembly during mitosis in embryos expressing GFP::LMN-1 WT, or GFP::LMN-1 8A in control conditions or upon partial inactivation of *plk-1* by RNAi (*Figure 3*).

In embryos expressing GFP::LMN-1 WT, the lamina was normally depolymerized at the metaphase-to-anaphase transition and was almost undetectable at anaphase (*Figure 3C* panels b-e, h-k, *Figure 3—video 1*), however, lamina dispersal was prevented upon *plk-1* inactivation (*Figure 3C*, panels n-q, t-w, *Figure 3—video 2*). Quantification of the GFP signal showed that the amount of GFP::LMN-1 at the NE remained elevated throughout mitosis upon *plk-1* inactivation (*Figure 3D*, *Figure 3—source data 1*). Consistent with previous observations (*Link et al., 2018*), lamina disassembly was also significantly delayed in embryos expressing the *gfp::lmn-1* 8A transgene (*Figure 3—video 3*) but this phenotype was greatly exacerbated upon *plk-1* inactivation (*Figure 3E*, compare

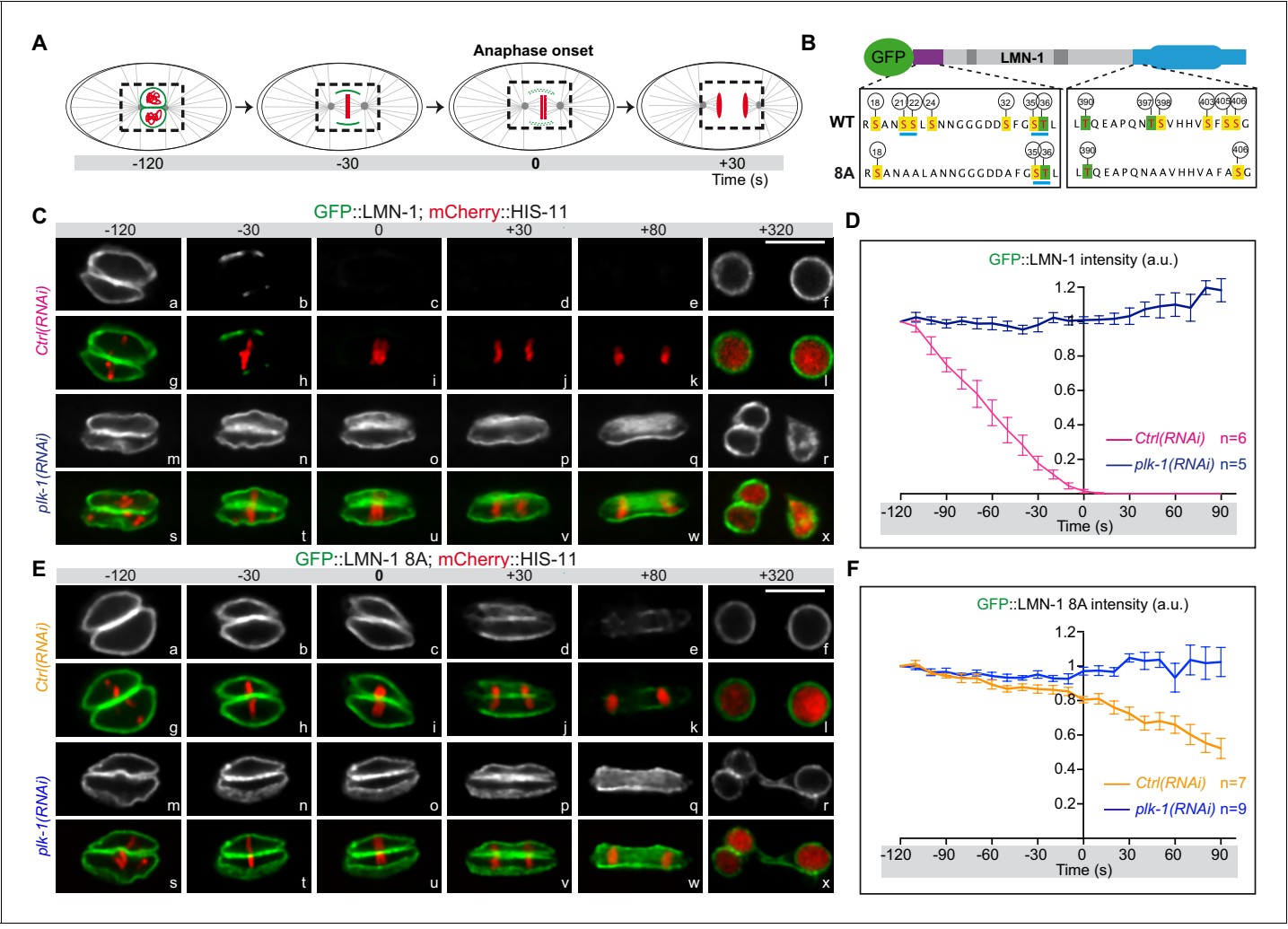

**Figure 3.** LMN-1-mediated PLK-1 phosphorylation promotes lamina disassembly during mitosis. (**A**) Schematics of the approach used to monitor and quantify lamina disassembly during mitosis using spinning disk confocal microscopy in embryos expressing GFP::LMN1 WT or 8A (green) and mCherry-HIS-11 (red). GFP::LMN-1 intensity was quantified over time in the boxed region relative to anaphase onset (time 0). (**B**) Schematics of GFP::LMN-1 wild-type and 8A with the position of the serine (boxed in yellow) and threonine (boxed in green) residues phosphorylated by PLK-1 in vitro. The eight serine residues substituted by non-phosphorylatable alanine in the head and the tail domains of GFP::LMN-1 8A are S21, S24, S22, S32, T397, T398, S403 and S405. The polo-docking sites are underlined in light blue. (**C**) Spinning disk confocal micrographs of *lmn-1Δ* mutant embryos expressing wild-type GFP::LMN-1 (shown alone, and in green in the merged images) and mCherry::HIS-11 (red, in the merged image) exposed to control or *plk-1(RNAi)*. Timings in second are relative to anaphase onset. All panels are at the same magnification. Scale Bar, 10 μm. D- Quantification of GFP::LMN-1 signal intensity above background at the NE in embryos of the indicated genotype during mitosis. The mean +/- SEM is presented for n = 6 embryos for control and n = 5 *plk-1(RNAi)* embryos. Data were collected from three independent experiments. (**E**) Spinning disk confocal micrographs of *lmn-1Δ* mutant embryos expressing GFP::LMN-1 8A (shown alone, and in green in the merged images) and mCherry::HIS-11 (red, in the merged image) exposed to control or *plk-1(RNAi)*. Timings, in seconds, are relative to anaphase onset. All panels are at the same magnification. Scale Bar, 10 μm. (**F**) Quantification of GFP::LMN-1 8A signal intensity above background at the NE in embryos of the indicated genotype during mitosis. The mean +/- SEM is presented for n = 7 embryos for control and n = 9 for *plk-1(RNAi)*. Data were collected from three independent experiments.

The online version of this article includes the following video, source data, and figure supplement(s) for figure 3:

**Source data 1.** related to *Figure 3D and F*.

**Figure supplement 1.** LMN-1 Serine 32 is phosphorylated in a PLK-1-dependent manner in one-cell *C. elegans* embryos.

**Figure 3—video 1.** Lamina disassembly in early embryos expressing GFP::LMN-1 and mCherry::HIS-11 exposed to control (video 1) or *plk-1(RNAi)* (video 2).

https://elifesciences.org/articles/59510#fig3video1

**Figure 3—video 2.** Lamina disassembly in early embryos expressing GFP::LMN-1 and mCherry::HIS-11 exposed to control (video 1) or *plk-1(RNAi)* (video 2).

https://elifesciences.org/articles/59510#fig3video2

*Figure 3 continued*
**Figure 3—video 3.** Lamina disassembly in early embryos expressing GFP::LMN-1 8A and mCherry::HIS-11 exposed to control (video 3) or *plk-1(RNAi)*
(video 4).
https://elifesciences.org/articles/59510#fig3video3
**Figure 3—video 4.** Lamina disassembly in early embryos expressing GFP::LMN-1 8A and mCherry::HIS-11 exposed to control (video 3) or *plk-1(RNAi)*
(video 4).
https://elifesciences.org/articles/59510#fig3video4

panels e-k with q-w, *Figure 3—video 4*). In *gfp::lmn-1* 8A mutant embryos, the lamina eventually disappeared in the vicinity of the centrosomes and between the parental chromosomes during anaphase (panels d-e, j-k) leading to two-cell embryos with a single nucleus in each blastomere at the two-cell stage (panels f-I, *Figure 3—video 3*). By contrast, upon *plk-1(RNAi)*, a dense meshwork of GFP::LMN-1 8A filaments encased the chromosomes throughout mitosis, systematically resulting in the formation of embryos with a paired nuclei phenotype (*Figure 3E*, panels r-x, *Figure 3—video 4*). Quantification of the GFP signal confirmed that GFP::LMN-1 8A levels at the NE did not decrease during mitosis upon *plk-1* inactivation as compared to control conditions (*Figure 3F*, *Figure 3—source data 1*). Taken together, these results indicate that PLK-1 phosphorylates LMN-1 to promote complete lamina disassembly during mitosis in early *C. elegans* embryos.

## *lmn-1* 8A mutant embryos display a penetrant paired nuclei phenotype

As PLK-1 phosphorylates numerous substrates during mitosis, the paired nuclei phenotype may be the direct or indirect result of failing to phosphorylate several PLK-1 targets. Alternatively, a defect in the depolymerization of the lamina meshwork might be sufficient to physically separate the parental chromosomes during mitosis, leading to the formation of 2 cell embryos with a paired nuclei phenotype. Supporting this latter hypothesis, we noticed that a small percentage of *gfp::lmn-1* 8A mutant embryos presented a double paired nuclei phenotype (8 out of 100, n = 100). This observation suggested that a greater stabilization of the lamina, using a stronger *lmn-1* gain-of-function allele, might be sufficient to separate the parental chromosomes during mitosis leading to a penetrant paired nuclei phenotype. We reasoned that the GFP tag in the N-terminal part of LMN-1 might destabilize the lamina filaments that assemble longitudinally in a head-to-tail fashion. We thus used CRISPR/Cas9 genome editing to mutagenize the *lmn-1* gene to generate and analyze a strain expressing untagged LMN-1 8A (see Materials and methods).

Remarkably, time-lapse differential interference contrast (DIC) microscopy showed that more than 65% of *lmn-1* 8A mutant embryos presented a penetrant paired nuclei phenotype at the two-cell stage (*Figure 4A and B*). Furthermore, almost 25% of these embryos died (*Figure 4C*, *Figure 4—source data 1*) illustrating the functional importance of depolymerizing the lamina in mitosis. LMN-1 immunostaining confirmed the persistence of the lamina between the parental chromosomes during mitosis in *lmn-1* 8A mutant embryos (*Figure 4D and E*). In wild-type embryos, the lamina disappears first in the vicinity of the centrosomes in pro-metaphase and from between the chromosomes during metaphase. By anaphase, LMN-1 is undetectable, reappearing only during NE reformation as the cells enter interphase at the two-cell stage (*Figure 4D*). By contrast, a strong lamina network persisted between parental chromosomes in *lmn-1* 8A mutant embryos throughout mitosis, eventually leading to the appearance of embryos with a paired nuclei phenotype (*Figure 4E*). These results unequivocally indicate that stabilization of the lamina, by preventing its phosphorylation by PLK-1, is sufficient to cause the appearance of embryos with a paired nuclei phenotype.

## Spatiotemporal PLK-1 localization at the NE follows the timing of lamina depolymerization

Having determined the role of PLK-1 in lamina depolymerization, we next analyzed PLK-1 spatiotemporal localization to the NE with respect to lamina depolymerization. (*Lee et al., 2000*). Before NEBD, PLK-1::sGFP is enriched in the vicinity of the centrosomes and between the parental chromosomes, precisely where the lamina is first depolymerized. PLK-1::sGFP is however undetectable on the NE located in the side of the pronuclei where the lamina persists until anaphase (*Figure 5A*, orange arrows). It is only right after the beginning of NEBD, manifested by indentation of the NE (white arrows), that PLK-1::sGFP starts to accumulate on the remnant of the envelope surrounding

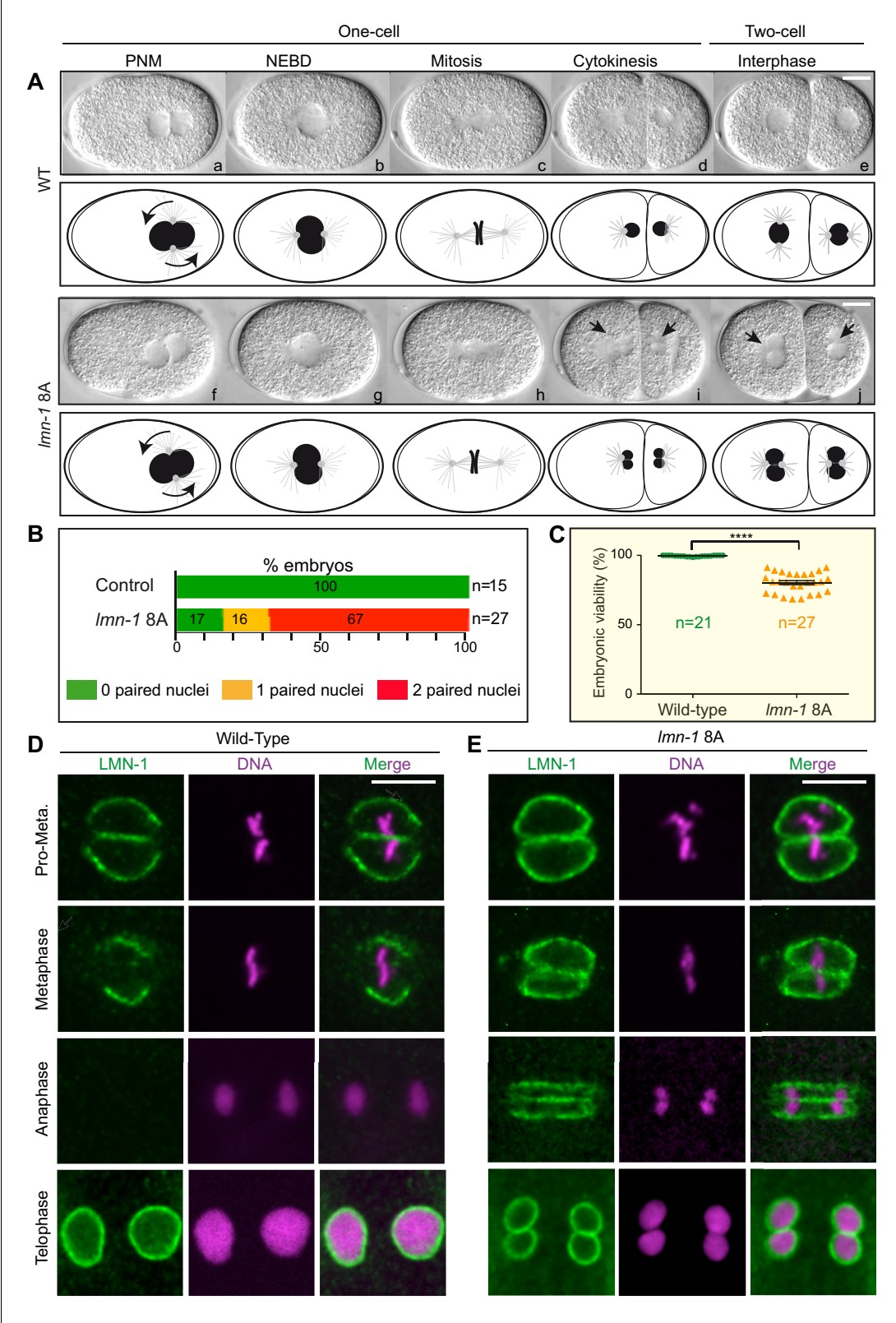

**Figure 4.** Non-phosphorylatable LMN-1 8A is sufficient to cause the appearance of embryos with a paired nuclei phenotype. (A) Differential interference contrast micrographs and corresponding schematics of the first two divisions in wild type (WT) (a–e) and *lmn-1 8A* (f–j) embryos at 23° C. Black arrowheads highlight paired nuclei in *lmn-1 8A* mutant embryos. PNM: Pronuclear meeting, NEBD: nuclear envelope breakdown. Scale Bar, 10 µm. (B) Percentage of WT and *lmn-1 8A* two-cell stage embryos presenting 0 (green bars), 1 (orange bars) or 2 (red bars) paired nuclei at the two-cell
*Figure 4 continued on next page*

*Figure 4 continued*

stage at 23°C. The number of embryos analyzed (n) is indicated on the right and was generated by aggregation over more than three independent experiments. The values inside the bars represent the percentage of embryos of a given phenotype. (C) Graph presenting the embryonic viability (%) determined from wild-type (n = 21) and *lmn-1 8A* (n = 27) mutant animals. Data were collected from two independent experiments containing each two replicates. The results are presented as means ± SEM. **** indicates p<0.0001. (D–E) Confocal images of fixed wild-type and *lmn-1 8A* mutant one-cell embryos in pro-metaphase, metaphase, anaphase and telophase, stained with LMN-1 antibodies (green) and counterstained with DAPI (magenta). All panels are at same magnification. Scale Bar, 10 µm.

The online version of this article includes the following source data for figure 4:

**Source data 1.** related to *Figure 4C*: Progeny test analysis of WT versus *lmn-1 8A* mutants.

the mitotic spindle where it accumulates during chromosome segregation (yellow arrows). These observations indicate that PLK-1::sGFP dynamics during mitosis closely follows the timing of lamina depolymerization, which is fully consistent with the lamina being disassembled by the PLK-1 kinase.

## PLK-1 kinase disassembles *C. elegans* LMN-1 filaments in vitro

The findings that PLK-1 directly phosphorylates LMN-1 and that expression of non-phophorylatable versions of LMN-1 prevent depolymerization of the lamina in vivo strongly suggest that PLK-1 phosphorylates LMN-1 to promote lamina depolymerization. To test directly this hypothesis, we established an in vitro assay for lamin depolymerization using pre-assembled LMN-1 filaments and PLK-1 kinase. Purified full-length *C. elegans* LMN-1 from *E. coli* (*Figure 5B* and Materials and methods) (*Foeger et al., 2006*) assembled into lamin filaments in vitro as confirmed by negative staining electron microscopy (*Figure 5C*). We next used a lamin pelleting assay as a readout of the solubility of the lamin before and after phosphorylation by PLK-1. In absence of PLK-1, in vitro assembled LMN-1 filaments were found in the pellet fraction after centrifugation (*Figure 5D*, panel a). After incubation with PLK-1 and ATP, LMN-1 was released into the supernatant fraction (*Figure 5D*, panel b). Moreover, the LMN-1 fraction recovered in the supernatant presented a slower migration on SDS-PAGE, likely indicative of phosphorylation. To corroborate these observations, we used electron microscopy to examine LMN-1 filaments 40 min after incubation with PLK-1. In the absence of PLK-1, a dense meshwork of LMN-1 filaments was detectable but this network was no longer visible in the presence of PLK-1 (*Figure 5—figure supplement 1*). Taken together, these results indicate that LMN-1 phosphorylation by PLK-1 promotes lamina disassembly in vitro.

## Non-phosphorylatable LMN-1 prevents the mixing of the parental chromosomes before the zygotic division

We next investigated whether stabilization of the lamina on its own is sufficient to physically separate the parental chromosomes during mitosis. Using embryos expressing a photoconvertible form of histone (Dendra2::H2B) (*Bolková and Lanctôt, 2016*), we photoconverted the female pronucleus (green-to-red) and followed the fate of the chromosomes during mitosis and at the 2 cell stage. In wild-type embryos, the maternal and paternal chromosomes, colored in red and green respectively, aligned on the metaphase plate and were found merged into a single nucleus at the two-cell stage after chromosome segregation (*Figure 6A and B*). By contrast, in *lmn-1* 8A mutant embryos, the parental chromosomes remained physically separated after chromosome segregation such that each of the paired nuclei received either the paternal or the maternal genome (*Figure 6B*). These observations indicate that a non-phosphorylatable version of LMN-1 is sufficient to physically separate the parental chromosomes during the first zygotic mitosis. These results also indicate that PLK-1, by phosphorylating LMN-1, promotes the merging of the parental genomes into a single nucleus after fertilization.

## Membrane gap formation between maternal and paternal pronuclei is absent in non-phosphorylatable *lmn-1* 8A mutant embryos

Merging of the parental chromosomes typically starts in metaphase with the formation of a membrane gap, which forms between maternal and paternal chromosomes coming into contact with the membrane of the pronuclei during chromosome congression and alignment on the metaphase plate (*Audhya et al., 2007*; *Rahman et al., 2015*). The timing of gap appearance suggests that it is coupled to the completion of metaphase plate alignment. This membrane gap is absent in *plk-1* mutant

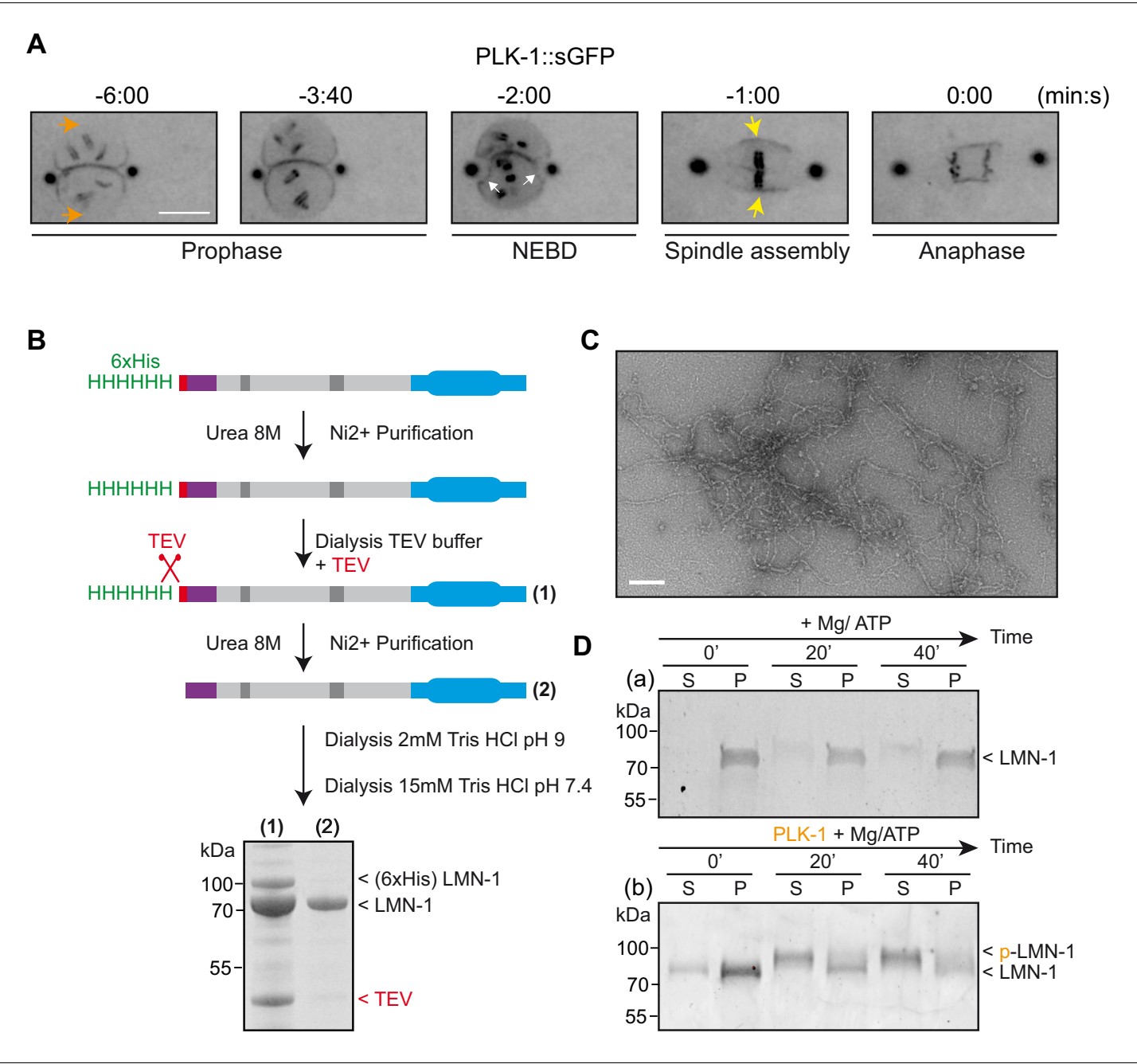

**Figure 5.** PLK-1 localizes at sites of lamina depolymerization in vivo and disassembles *C.e* lamin filaments in vitro. (**A**) Images from a time-lapse spinning disk confocal movie showing the spatiotemporal localization of PLK-1::sGFP to centrosomes, chromosomes and to the NE during the first mitosis of the *C. elegans* zygote. Orange arrows point to the regions of the NE located in the regions of the pronuclei where PLK-1::sGFP is excluded before NEBD. NEBD is defined as the time point at which the nuclear envelope starts to deform (white arrows). Yellow arrows point to the region of the NE surrounding the mitotic spindle where PLK-1::sGFP accumulates. Times are in minutes:seconds relative to anaphase onset. Scale bar, 5 μm. (**B**) Flow-chart of the approach used to purify LMN-1 from *E. coli* and to assemble LMN-1 filaments in vitro. Coomassie blue staining of purified 6xHis-LMN-1 before and after cleavage of the 6xHis tag with TEV protease. As the TEV protease is also tagged with the 6xHis tag (not mentioned in the figure), the protease is removed from the LMN-1 preparation after purification on Ni²⁺ column (step 2). (**C**) Negative staining electron microscopy micrograph of *C. elegans* lamin filaments assembled from purified LMN-1. Scale Bar, 100 nm. **D-** In vitro kinase assay was performed with Mg/ATP alone as control (**a**) or with PLK-1 plus Mg/ATP (**b**) and LMN-1 filaments as substrate. The samples were taken at the indicated time, centrifuged and the pellet (P) and supernatant (S) fractions were subjected to SDS-PAGE and the gel was stained with Coomassie brilliant blue.

The online version of this article includes the following figure supplement(s) for figure 5:

**Figure supplement 1.** PLK-1 kinase promotes LMN-1 filaments disassembly in vitro.

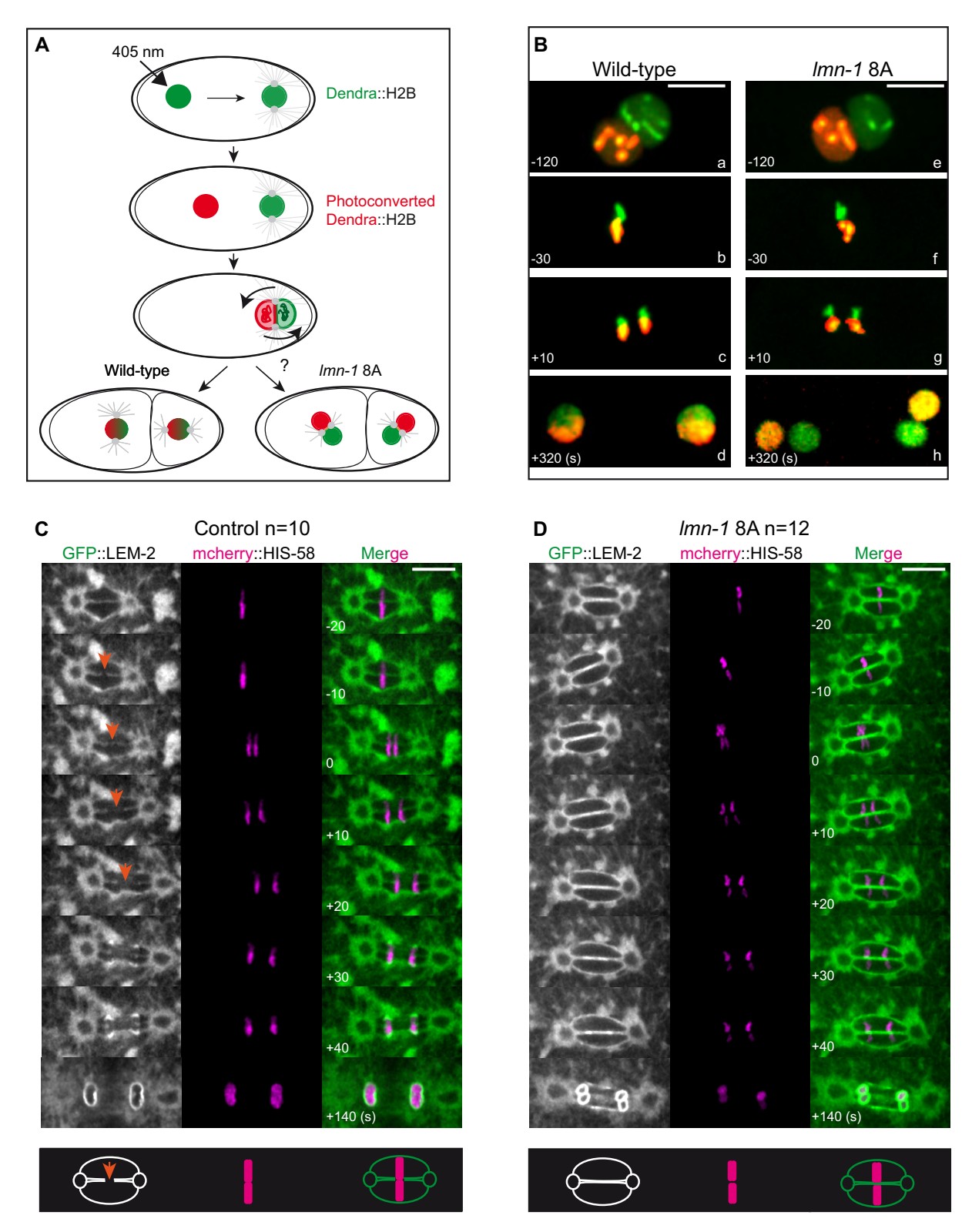

**Figure 6.** Paternally- and maternally-derived chromatin remain physically separated during mitosis in *lmn-1 8A* mutant one-cell embryos. (**A**) Schematics of the live-imaging approach used to differentially label parental chromosomes. The female pronuclei of the Dendra2-H2B-expressing strain are photoconverted (405 nm laser beam) in the early zygote changing the fluorescence of the female pronucleus from green-to-red and the embryos are imaged until the 2 cell stage. (**B**) Representative spinning disk confocal images are taken from different wild-type (**a, d**) and *lmn-1 8A* (**e, f**) mutant

*Figure 6 continued on next page*

*Figure 6 continued*

embryos. Shown are maximum projections of image stacks. Nuclei are in mitotic prophase (**a, e**), metaphase (**b, f**) anaphase (**c, g**) and telophase (**d, h**). Scale Bar, 10 μm. (**C-D**) Spinning disk confocal micrographs of WT and *lmn-1 8A* mutant one-cell stage embryos expressing the INM protein LEM-2 fused to GFP (GFP::LEM-2, shown alone on the right of the panels and in green in the merged images) and mCherry::HIS-58 (magenta). In the control embryo (left), the nuclear envelope breaks as the chromosomes align at the metaphase plate (orange arrows), allowing the chromosomes from the sperm and oocyte pronuclei to merge on the spindle before their segregation. In the *lmn-1 8A* mutant embryos, the nuclear envelope remains intact and acts as a physical barrier that separates the chromosomes from the two pronuclei during their segregation. On the bottom panels, schematics illustrate the presence of a membrane gap in wild-type as opposed to *lmn-1 8A* embryos. Scale Bar, 10 μm. Times are in seconds relative to anaphase onset.

The online version of this article includes the following video and figure supplement(s) for figure 6:

**Figure supplement 1.** *lmn-1 8A* mutant one-cell embryos are defective in membrane gap formation but properly align parental chromosomes.

**Figure 6—video 1.** Control (video 1) or *lmn-1 8A* (video 2) mutant embryos expressing GFP::LEM-2 and mCherry::HIS-58.

https://elifesciences.org/articles/59510#fig6video1

**Figure 6—video 2.** Control (video 1) or *lmn-1 8A* (video 2) mutant embryos expressing GFP::LEM-2 and mCherry::HIS-58.

https://elifesciences.org/articles/59510#fig6video2

embryos (*Rahman et al., 2015*). However, because chromosome congression and alignment are impaired in *plk-1* mutant embryos, a defect in membrane gap formation might be an indirect consequence of this congression defect rather than of failure to phosphorylate the lamins.

To investigate this possibility, we used spinning disk confocal microscopy to examine the configuration of chromosomes and the NE in wild type or *lmn-1 8A* mutant embryos expressing mCherry::HIS-58 and the inner nuclear membrane protein LEM-2::GFP. In wild-type one-cell embryos, chromosome congression and alignment on the metaphase plate are coincident with the appearance of a membrane scission event in the NE between the two pronuclei (*Figure 6C*, orange arrows, *Figure 6—video 1*), consistent with previous observations (*Audhya et al., 2007*). This scission event allows the chromosomes from the two pronuclei to mingle on the metaphase plate and thus to merge into a single nucleus after chromosome segregation. However, this scission event of the membrane was never seen in metaphase *lmn-1* 8A mutant embryos (n = 12) (*Figure 6D*, *Figure 6—video 2*), even though parental chromosomes properly congressed and aligned on the metaphase plate in a large fraction of embryos (7 out of 12 embryos) (*Figure 6—figure supplement 1A,B*). A persisting membrane barrier was maintained in these embryos such that parental chromosomes segregated as four, instead of two DNA masses. Based on these observations, we conclude that PLK-1-mediated lamina depolymerization is a prerequisite for membrane gap formation and ultimately, for the merging of the parental chromosomes into a single nucleus at the two-cell stage.

## Discussion

The *C. elegans* zygote provides a dynamic developmental context to study the mechanisms by which parental chromosomes are merged at the beginning of life. In *C. elegans*, the sperm brings in the centrosomes (*Albertson, 1984*), similar to the situation in humans and unlike the situation in mouse, where the first mitotic division is acentrosomal (*Sathananthan et al., 1991*) with parental chromosomes aligning on two separate mitotic spindles (*Reichmann et al., 2018*). Thus, after fertilization, the *C. elegans* embryo, and likely also the human embryo, must align and mingle the parental chromosomes on a single mitotic spindle. For this event to occur properly, the nuclear envelopes surrounding the egg and sperm pronuclei have to be removed and/or reorganized in a timely fashion (*Rahman et al., 2020*). Here we show that PLK-1 triggers lamina depolymerization in mitosis, which is essential to merge the parental chromosomes in a single nucleus at the two-cell stage (*Figure 7*). A *lmn-1* version carrying eight phosphosite mutations is sufficient to prevent membrane scission between pronuclei and the merging of the parental chromosomes. Our findings thus indicate that lamina depolymerization is a key step allowing the merging of parental chromosomes at the beginning of *C. elegans*, and possibly also of human life.

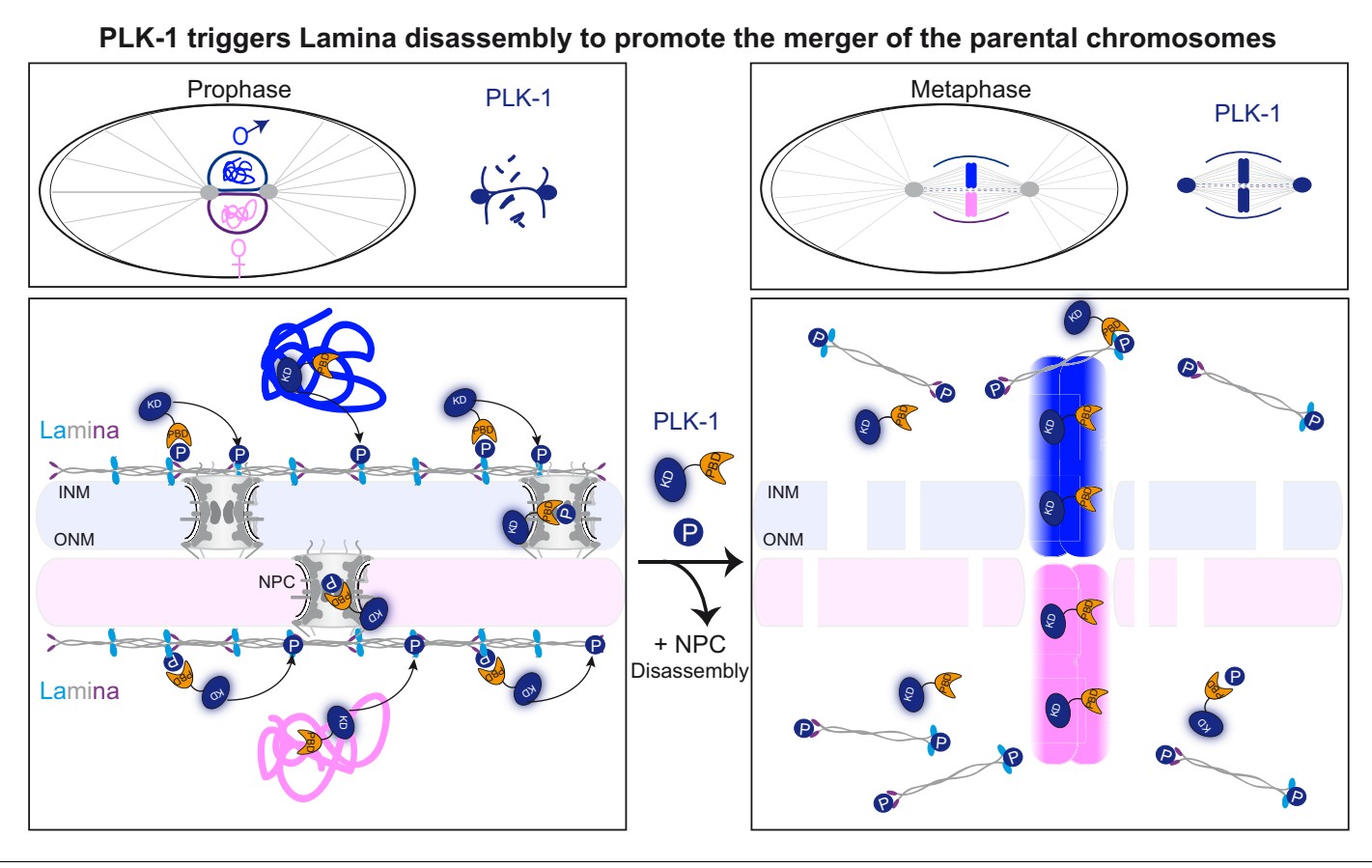

**Figure 7.** PLK-1 dynamics at the nuclear envelope dictates the timing of lamina depolymerization during mitosis The maternal and paternal chromosomes are colored in pink and blue respectively. Polo-like kinase1 with the kinase domain (KD) in dark blue and the polo-box domain (PBD) in orange. Phosphorylated residues (P) circled in dark blue are modified by PLK-1.

## PLK-1 promotes the merger of the parental genomes into a single nucleus after fertilization by triggering lamina depolymerization

After fertilization, the male and female pronuclei, each surrounded by a nuclear envelope have to fuse to allow the merging of the parental chromosomes. Like all NEs, each pronucleus NE is a double membrane perforated by nuclear pore complexes (NPCs). The outer nuclear membrane is continuous with the ER while the inner nuclear membrane is supported by nuclear lamina, which provides structural integrity to the pronuclei (*Cohen-Fix and Askjaer, 2017*). Understanding how these double membranes are remodeled to allow the merging of the parental chromosomes is of critical importance. Although it is a key step in development, this process is poorly understood.

How does PLK-1 promote NEBD and pronuclear membrane fusion? Rahman et al have recently used ion beam scanning electron microscopy (FIB-SEM) to show that the membranes of the two pronuclei fuse during metaphase and become two membranes perforated by fenestrations, allowing the parental chromosomes to come into contact. The pronuclei membranes fuse by two types of membrane structures: outer-outer junctions and three-way sheet junctions (*Rahman et al., 2020*). In *plk-1*ts embryos, these three-way junctions are not observed and the membranes do not fuse (*Rahman et al., 2020*). Based on these observations, it has been proposed that PLK-1 promotes the fusion of the pronuclei membranes (*Rahman et al., 2020*) and that membrane fusion events, rather than NEBD, drive the two pronuclei together before the first mitosis (*Ma and Starr, 2020*). While we cannot exclude a direct role of PLK-1 in pronuclei membrane fusion, our results rather suggest that PLK-1 is required to depolymerize the lamina and possibly to remove other nuclear envelope proteins (nucleoporins, inner nuclear membrane proteins) from the nuclear envelopes of the pronuclei, to promote the merging of the parental chromosomes. Expression of LMN-1 mutated on eight PLK-

1-phosphorylation sites is sufficient to stabilize the lamina during mitosis, which is in turn sufficient to physically separate the parental chromosomes during their segregation in anaphase, giving rise to embryos with a paired nuclei phenotype. Furthermore, the membrane scission event, required for chromosome merging in metaphase, was never seen in *lmn-1* 8A mutant embryos, even though parental chromosomes perfectly aligned on the metaphase plate in a majority of embryos (7 out of 12). These findings suggest that PLK-1-mediated lamina depolymerization, a key step of NEBD (*Güttinger et al., 2009*), is a prerequisite for membrane gap formation, pronuclei membrane fusion, and for the merging of the parental chromosomes into a single nucleus at the two-cell stage. Notably, embryonic lethality was associated with the *lmn-1* 8A mutant, illustrating the functional importance of lamina depolymerization. The cause of embryonic lethality is unclear but might result from defects in chromosome movements in meiosis (*Link et al., 2018*) and segregation in mitosis. A similar defect in the merging of parental chromosomes at the beginning of human life might have drastic consequences and cause various defects including loss of heterozygosity.

It is worth mentioning that although PLK-1 phosphorylates LMN-1 on fifteen sites, here we have only studied the functional consequences of mutating eight of these sites. We clearly showed that mutation of eight sites is sufficient to stabilize the lamina during mitosis but does not fully recapitulate the stabilization of the lamina observed after *plk-1* depletion. We attributed this difference to the seven remaining sites but we can certainly not exclude the possibility that PLK-1 also phosphorylates other Lamin-binding proteins which indirectly contributes to lamina disassembly. To further investigate this issue, it will be critical to analyze lamina disassembly in embryos expressing GFP::LMN-1 15A. Likewise, it will be important to decipher whether all these sites equally contribute to lamina disassembly and whether they are equally phosphorylated by PLK-1. Addressing this issue might help to decipher how lamina disassembly is spatially regulated in the early embryo. Our study also revealed that N-terminal tagging of LMN-1 with GFP appears to alter the assembly properties of the *C. elegans* lamina. Clearly the phenotype observed after expression of GFP::LMN-1 8A is not as severe as the one resulting from the expression of untagged LMN-1 8A. Whether GFP has the same impact on lamins from other species is unclear but it will be important to carefully examine this possibility in the future.

## Possible contribution of PLK-1 to lamina depolymerization in other species

Depolymerization of the lamina during mitosis is a major step in NEBD (*Gerace and Blobel, 1980*). Seminal works have shown the importance of the master mitotic Cyclin B-Cdk1 kinase in this process in human cells (*Peter et al., 1990*; *Ward and Kirschner, 1990*; *Heald and McKeon, 1990*; *Machowska et al., 2015*). Although the importance of Cyclin B-Cdk1-mediated phosphorylation of lamins for their depolymerization during mitosis cannot be overemphasized, several questions remain to be addressed. For instance, lamin A/C and B networks are disassembled with different kinetics (*Georgatos et al., 1997*) but the precise contribution of Cyclin B-Cdk1 in the depolymerization of both networks is not entirely clear. While substitution of CDK sites by alanines has been shown to strongly stabilize lamin A filaments in human cells (*Heald and McKeon, 1990*), similar point mutations in lamin B1 have only a modest effect on the timing of lamin B1 depolymerization during mitosis (*Mall et al., 2012*).

Another important aspect to consider is the timing of lamina depolymerization in different species. If lamina depolymerization occurs in prophase and pro-metaphase in human cells that undergo 'open mitosis' (*De Souza and Osmani, 2009*), it occurs at the latter stages in several organisms including *Drosophila melanogaster* or *C. elegans.* In these organisms, part of lamina dispersion takes place at the metaphase-to-anaphase transition (*Paddy et al., 1996*; *Lee et al., 2000*) and coincides with a drop in Cyclin B-Cdk1 kinase activity due to proteasomal Cyclin B degradation (*van der Voet et al., 2009*). This suggests that other kinase(s) might be responsible for lamina depolymerization in these organisms at this cell cycle stage. Cyclin B-Cdk1 might initiate the process and another kinase might act after Cyclin B-Cdk1 inactivation, or alternatively, the process might be entirely CDK-independent. Our results strongly suggest that lamina depolymerization in *C. elegans* is independent of Cyclin-Cdk and is driven by the Polo-like kinase PLK-1. In *Drosophila*, CDK1 is directly implicated but Polo might also play a role. In support of this hypothesis, we found that PLK-1 binds *Drosophila* head domain of lamin Dm0 via non-self-priming and binding mechanisms. Furthermore, previous work established that Polo is required for NEBD in *Drosophila* (*Kachaner et al., 2017*) and that lamin

Dm0 phosphorylation outside the CDK site contributes to lamin dispersal in mitosis (*Mehsen et al., 2018*). Based on all these observations, and given the timing of lamina disassembly in *Drosophila*, we suggest that Cyclin-Cdk, together with Polo, might ensure the efficient depolymerization of lamin in mitosis. Likewise, Plk1 could assist lamin A/C dispersal early in mitosis in human cells because similarly to *Drosophila* lamin Dm0, Lamin A/C is readily phosphorylated on a polo-docking site in mitosis (*Dephoure et al., 2008*; *Chen et al., 2013*). Further work will be required to test the role of Plk1 in lamina depolymerization in other organisms.

In conclusion, we have identified a PLK-1-dependent lamina-depolymerization process in the *C. elegans* early embryo that is essential for parental chromosome mixing after the first mitosis. Given that human embryos presumably also segregate parental chromosomes on a single mitotic spindle, timely lamina depolymerization might be required in human embryos to ensure parental chromosomes mixing. A defect in this process might be deleterious for human embryonic development, leading for example to the production of genomic heterogeneity among the blastomeres. It will be important to consider defects in the merging of the parental chromosomes in the fertilized zygote as a source of multinucleated blastomeres.

# Materials and methods

**Key resources table**

| Reagent type (species) or resource | Designation | Source or reference | Identifiers | Additional information |
|---|---|---|---|---|
| Strain, strain background (*Caenorhabditis elegans*) | *C. elegans* N2 Bristol | CGC | http://www.cgc.cbs.umn.edu/strain.php?id=10570 | |
| Strain, strain background (*C. elegans*) | [*gfp::lmn-1*] MosSCI: lmn-1(tm1502) I; jfSi68[Plmn-1::gfp cb-unc-119(+)] II | *Link et al., 2018*, #49005 | UV120 | |
| Strain, strain background (*C. elegans*) | [*gfp::lmn-1$^{S8A}$*] MosSCI: lmn-1(tm1502) I; jfSi89[Plmn-1S(21,22,24,32,397, 398,403,405)A::gfp cb-unc-119(+)] II | *Link et al., 2018*, #49005 | UV122 | |
| Strain, strain background (*C. elegans*) | lmn-1(jf140 [S21,22,24,32,397,398, 403,405A]) I/hT2 [bli-4(e937) let-?(q782) qIs48] (I;III) | This study | UV2059 | V. Jantsch Lab |
| Strain, strain background (*C. elegans*) | tonSi1[mex-5p::Dendra2::his-66::tbb-2 3'UTR + Cbr-unc-119(+)]II | *Bolková and Lanctôt, 2016*, #90027 | JBL1 | |
| Strain, strain background (*C. elegans*) | lmn-1(jf140 [S21,22,24,32,397,398, 403,405A]) I/hT2 [bli-4(e937) let-?(q782) qIs48] (I;III);tonSi1 [mex-5p::Dendra2::his-66::tbb-2 3'UTR + Cbr-unc-119(+)]II | This study | WLP831 | L. Pintard Lab |
| Strain, strain background (*C. elegans*) | Itls37 [(pAA64) pie-1p::mCherry::his-58 + unc-119(+)]IV qals3507 [pie-1::GFP::LEM-2 + unc-119(+)] | CGC | OD83 | |
| Strain, strain background (*C. elegans*) | lmn-1$^{S8A}$S (21,22,24,32,397,398,403, 405)A Itls37 [(pAA64) pie-1p::mCherry::his-58 + unc-119(+)]IV qals3507 [pie-1::GFP::LEM-2 + unc-119(+)] | This study | WLP833 | L. Pintard Lab |

*Continued on next page*

*Continued*

| Reagent type (species) or resource | Designation | Source or reference | Identifiers | Additional information |
|---|---|---|---|---|
| Strain, strain background (*C. elegans*) | *plk-1(lt17)* ([*plk-1::sgfp*] *loxp*)III | *Martino et al., 2017*, #66043 | OD2425 | |
| Strain, strain background (*Escherichia coli*) | BL21(DE3) | Sigma-Aldrich | CMC0016 | |
| Antibody | LMN-1 (Rabbit polyclonal) | Novus Biologicals | Cat#38530002 RRID:AB_10005072 | (1:100 IF) (1:1,000 WB) |
| Antibody | LMN-1Ser32pi (Rabbit polyclonal) | *Link et al., 2018*, #49005 | V. Jantsch Lab (Home made) | (1:10 IF) |
| Antibody | Plk1 (human) (Mouse monoclonal) | Merck Millipore | Cat#05–844 RRID:AB_310836 | (1:1,000 WB) |
| Antibody | GST (Rabbit polyclonal) | This study | L. Pintard Lab (Home made) | (1:1,000 WB) |
| Antibody | Anti-Mouse IgG (Fab specific) -Peroxidase antibody (produced in goat) | Sigma | Cat#A9917 RRID:AB_258476 | (1:5,000 WB) |
| Antibody | Anti-Rabbit IgG (whole molecule) -Peroxidase antibody (Produced in goat) | Sigma | Cat#A0545 RRID:AB_257896 | (1:3,000 WB) |
| Antibody | Anti-Rabbit IgG (H+L) Cross-Adsorbed Secondary Antibody, Alexa Fluor 568 (Produced in goat) Fluor 568 | Invitrogen | Cat#A-11011 RRID:AB_143157 | (1:1,000 IF) |
| Antibody | Anti-Mouse IgG (H+L) Cross-Adsorbed Secondary Antibody, Alexa Fluor 488 (Produced in goat) | Invitrogen | Cat#A-11001 RRID:AB_2534069 | (1:1,000 IF) |
| Chemical compound, drug | Coomassie R250 | Sigma | Cat#B014925G | |
| Chemical compound, drug | Ponceau Red | Sigma | Cat#A1405 | |
| Chemical compound, drug | VECTASHIELD Mounting Medium with DAPI | Eurobio | Cat#H-1200 | |
| Chemical compound, drug | IPTG | Euromedex | Cat#EU0008-B | |
| Chemical compound, drug | Adenosine TriPhosphate (ATP) | Sigma | Cat#A2383 | |
| Chemical compound, drug | $\gamma$-[$^{32}$P] Adenosine TriPhosphate (ATP) 500 $\mu$Ci 3000 Ci/mmol | PerkinElmer | Cat#NEG002A500U | |
| Chemical compound, drug | Phos-Tag Acrylamide AAL-107 | WAKO-SOBIODA | Cat#W1W304-93521 | |
| Chemical compound, drug | Imidazole | Sigma | Cat#I202 | |
| Chemical compound, drug | Glutathione | Sigma | Cat#G4251 | |
| Chemical compound, drug | HiTrap Chelating HP 5 $\times$ 1 mL | GE Healthcare | Cat#17-0408-01 | |

*Continued*

| Reagent type (species) or resource | Designation | Source or reference | Identifiers | Additional information |
|---|---|---|---|---|
| Chemical compound, drug | Glutathion Sepharose 4B Fast Flow | GE Healthcare | Cat#17-0756-01 | |
| Chemical compound, drug | Pfu | Promega | Cat#M7741 | |
| Chemical compound, drug | DpnI | Biolabs | Cat#R0176S | |
| Commercial assay or kit | ECL reagent | Millipore | Cat#WBKLS0500 | |
| Commercial assay or kit | BP Clonase II Enzyme Mix (Gateway cloning) | Invitrogen | Cat#11789–020 | |
| Commercial assay or kit | LR Clonase II Enzyme Mix (Gateway cloning) | Invitrogen | Cat#11791–020 | |
| Recombinant protein | Human Cyclin B-Cdk1 kinase | New England Biolabs | Cat#P6020L | |
| Recombinant DNA reagent | L4440 (RNAi Feeding vector) | *Kamath et al., 2001*, #2201 | N/A | |
| Recombinant DNA reagent | *plk-1* cloned into L4440 | *Kamath et al., 2003*, #46063 | Arhinger Library | |
| Recombinant DNA reagent | pDESTttTi5605[R4-R3] for MOS insertion on Chromosome II | *Frøkjaer-Jensen et al., 2008*, #79702 | pCFJ150 Addgene plasmid # 19329 | |
| Recombinant DNA reagent | MOS transposase Pglh-2::MosTase::glh-2utr | *Frøkjaer-Jensen et al., 2008*, #79702 | pJL43.1 Addgene plasmid # 19332 | |
| Recombinant DNA reagent | *Prab-3::mCherry* | *Frøkjaer-Jensen et al., 2008*, #79702 | pGH8 | |
| Recombinant DNA reagent | *Pmyo-2::mCherry::unc-54* | *Frøkjaer-Jensen et al., 2008*, #79702 | pCFJ90 | |
| Recombinant DNA reagent | Entry clone for Gateway pDONR221 (*lmn-1* CDS was cloned in this vector) | Multistite Gateway Kit, Thermo Fischer Scientific | Cat# 12537–023 | |
| Recombinant DNA reagent | pDONR P4-P1R (the *lmn-1* 5' UTR with and without GFP was cloned in this vector) | Multistite Gateway Kit, Thermo Fischer Scientific | Cat# 12537–023 | |
| Recombinant DNA reagent | pDONRTM P2r-P3 (the *lmn-1* 3'UTR was cloned in this vector) | Multistite Gateway Kit, Thermo Fischer Scientific | Cat# 12537–023 | |
| Recombinant DNA reagent | Minigene from IDT for generating the lmn-18A containing the whole CDS of *lmn-1*(S21,22,24,32,397, 398,403,405A) | *Link et al., 2018*, #49005 | N/A | |
| Recombinant DNA reagent | Plmn-1_gfp::lmn-1_ lmn-1 3'UTR in pCFJ150 | *Link et al., 2018*, #49005 | N/A | |
| Recombinant DNA reagent | Plmn-1_gfp::lmn-1(S8A) lmn-1 3'UTR in pCFJ150 | *Link et al., 2018*, #49005 | N/A | |
| Recombinant DNA reagent | Gal4 pDEST DB | *Noatynska et al., 2010*, #23221 | pMG97 | |
| Recombinant DNA reagent | Gal4 pDEST DB-PLK-1 PBD | *Noatynska et al., 2010*, #23221 | pMG477 | |
| Recombinant DNA reagent | Gal4 pDEST DB-PLK-1 PBD H542A, K544M | *Noatynska et al., 2010*, #23221 | pMG538 | |
| Recombinant DNA reagent | pGEX-4T (GST) | GE Healthcare | Cat#GE28-9545-49 | |

*Continued on next page*

*Continued*

| Reagent type (species) or resource | Designation | Source or reference | Identifiers | Additional information |
|---|---|---|---|---|
| Recombinant DNA reagent | pFasTBAC Hta PLK-1 *C. elegans* | *Tavernier et al., 2015*, #22388 | pLP871 | |
| Recombinant DNA reagent | pGEX-6p1 GST-PLK-1 PBD *H. s* | Gift I. Sumara | N/A | |
| Recombinant DNA reagent | pGEX-6p1 GST-PLK-1 PBD H538A/K540M *H. s* | Gift I. Sumara | N/A | |
| Recombinant DNA reagent | pDONR201 TEV-Aurora A *H. s* | This study | pLP1884 | |
| Recombinant DNA reagent | His-GST-TEV-Aurora A *H. s* | This study | pLP2067 | |
| Recombinant DNA reagent | pBABE-puro-GFP-wt-lamin A | Gift B. Cisneros | Addgene #17662 | |
| Recombinant DNA reagent | cDNA LMN-1 13A (S18A, S21A, S22A, S24A, S32A, S35A, T36A, T390A, T397A, S398A, S403A, S405A, S406A) Generate by Thermofisher | ThermoFisher | N/A | |
| Recombinant DNA reagent | pDONR201 | ThermoFisher | N/A | |
| Recombinant DNA reagent | pDONR201 LMN-1 [H] (aa1-47) | This study | pLP2234 | |
| Recombinant DNA reagent | pDONR201 LMN-1 [H] (aa1-47) 3A S18A S35A T36A | This study | pLP2277 | |
| Recombinant DNA reagent | pDONR201 LMN-1 [H] (aa1-47) 4A S21A S22A S24A S32A | This study | pLP2236 | |
| Recombinant DNA reagent | pDONR201 LMN-1 [H] (aa1-47) 7A S18A S21A S22A S24A S32A S35A T36A | This study | pLP2276 | |
| Recombinant DNA reagent | pDONR201 LMN-1 [T] (aa387-548) | This study | pLP2270 | |
| Recombinant DNA reagent | pDONR201 LMN-1 [T1] (aa387-436) | This study | pLP2249 | |
| Recombinant DNA reagent | pDONR201 LMN-1 [T1] 4A (aa387-436) T397A S398A S403A S405A | This study | pLP2250 | |
| Recombinant DNA reagent | pDonR201 LMN-1 [T1] 6A (aa387-436) T390A, T397A, S398A, S403A, S405A, S406A | This study | pLP2269 | |
| Recombinant DNA reagent | pDONR201 cDNA of lamin Dm0 from *Drosophila melanogaster* | *Mehsen et al., 2018*, #44350 | N/A | |
| Recombinant DNA reagent | pDONR201 cDNA of lamin with 7 mutations of the CDK motifs from *Drosophila melanogaster* T12A, T20A, S42A, S45A, T413A, T435A, T440A | *Mehsen et al., 2018*, #44350 | N/A | |
| Recombinant DNA reagent | pDONR201 Lamin Dm0 [H] (aa1-57) | This study | pLP2314 | |
| Recombinant DNA reagent | pDONR201 Lamin Dm0 [H] (aa1-57) 4A T12A T20A S42A S44A | This study | pLP2315 | |
| Recombinant DNA reagent | pDONR201 Lamin A/C [H] (aa1-34) | This study | pLP2316 | |

*Continued on next page*

*Continued*

| Reagent type (species) or resource | Designation | Source or reference | Identifiers | Additional information |
|---|---|---|---|---|
| Recombinant DNA reagent | pDONR201 TEV-Bora 1–224 *Hs* | This study | pLP1848 | |
| Recombinant DNA reagent | pDEST17 His-TEV-Bora 1–224 *Hs* | This study | pLP1850 | |
| Recombinant DNA reagent | pDEST15 | ThermoFisher | Cat#11802014 | |
| Recombinant DNA reagent | pDEST15 GST-LMN-1 [H] (aa 1–47) | This study | pLP2235 | |
| Recombinant DNA reagent | pDEST15 GST- LMN-1 [H] (aa 1–47) 3A S18A S35A T36A | This study | pLP2282 | |
| Recombinant DNA reagent | pDEST15 GST- LMN-1 [H] (aa 1–47) 4A S21A S22A S24A S32A | This study | pLP2237 | |
| Recombinant DNA reagent | pDEST15 GST- LMN-1 [H] (aa1-47) 7A S18A S21A S22A S24A S32A S35A T36A | This study | pLP2281 | |
| Recombinant DNA reagent | pDEST15 GST- LMN-1 [H] (aa1-47) 9A S18A S21A S22A S24A S32A S35A T36A T40A S41A | This study | pLP2393 | |
| Recombinant DNA reagent | pDEST15 GST- LMN-1 [T] (aa387-548) | This study | pLP2271 | |
| Recombinant DNA reagent | pDEST15 GST- LMN-1 [T1] (aa387-436) | This study | pLP2253 | |
| Recombinant DNA reagent | pDEST15 GST- LMN-1 [T1] 4A (aa387-436) T397A S398A S403A S405A | This study | pLP2254 | |
| Recombinant DNA reagent | pDEST15 GST- LMN-1 [T1] 6A (aa387-436) T390A, T397A, S398A, S403A, S405A, S406A | This study | pLP2298 | |
| Recombinant DNA reagent | pDEST15 GST- Lamin Dm0 [H] (aa1-57) | This study | pLP2322 | |
| Recombinant DNA reagent | pDEST15 GST- Lamin Dm0 [H] (aa1-57) 4A T12A T20A S42A S44A | This study | pLP2323 | |
| Recombinant DNA reagent | pDEST15 GST- Lamin A/C [H] (aa1-34) | This study | pLP2324 | |
| Recombinant DNA reagent | pDEST15 GST- Lamin A/C [H] (aa1-34)1A T18A | This study | pLP2362 | |
| Recombinant DNA reagent | pET28b | Novagen | Cat#69864 | |
| Recombinant DNA reagent | pET28b (His linker-TEV-linker Nter) LMN-1 | This study | pLP2231 | |
| Recombinant DNA reagent | Alt-R CRISPR-Cas9 tracrRNA, 100 nmol | IDT | Cat#1072534 | |
| Sequence-based reagent | Primers for cloning and site-directed mutagenesis (see oligonucleotide sequences table S4 source) | This study | N/A | |
| Software, algorithm | Clustal Omega | EMBL-EBI | https://www.ebi.ac.uk/Tools/msa/clustalo/ | |
| Software, algorithm | Jalview | *Waterhouse et al., 2009*, #45561 | https://www.jalview.org/ | |
| Software, algorithm | Adobe Illustrator CS6 | Adobe | https://www.adobe.com/products/illustrator.html | |

*Continued*

| Reagent type (species) or resource | Designation | Source or reference | Identifiers | Additional information |
|---|---|---|---|---|
| Software, algorithm | Adobe Photoshop CS4 | Adobe | https://www.adobe.com/products/photoshop.html | |
| Software, algorithm | Image J | NIH; *Schneider et al., 2012*, #74926 | https://imagej.nih.gov/ij/ | |
| Software, algorithm | ZEN | Zeiss | https://www.zeiss.com/microscopy/int/products/microscope-software/zen.html | |
| Software, algorithm | PRISM | Graphpad | https://www.graphpad.com/ | |
| Software, algorithm | Metamorph | Molecular Devices | https://www.metamorph.com/ | |
| Software, algorithm | Proteome Discoverer 2.2 | Thermo Scientific | https://www.thermofisher.com/store/products/OPTON-30945#/OPTON-30945 | |
| Software, algorithm | Matrix Science 5.1 | Mascot Server | https://www.matrixscience.com/ | |

## Contact for reagent and resource sharing

Further information and requests for reagents may be directed to and will be fulfilled by the lead contact author L. Pintard lionel.pintard@ijm.fr.

## Experimental model and subject details

*C. elegans* and bacterial strains used in this study are listed in the Key Resources Table.

## Method details

### Molecular biology

The plasmids and oligonucleotides used in this study are listed in the Key resource table. Gateway cloning was performed according to the manufacturer's instructions (Invitrogen). All the constructs were verified by DNA sequencing (GATC-Biotech).

### Nematode strains and RNAi

*C. elegans* strains were cultured and maintained using standard procedures (*Brenner, 1974*). RNAi was performed by the feeding method using HT115 bacteria essentially as described (*Kamath et al., 2001*), except that 2 mM of IPTG was added to the NGM plates and in the bacterial culture just prior seeding the bacteria. As a control, animals were exposed to HT115 bacteria harboring the empty feeding vector L4440 (mock RNAi). RNAi clones were obtained from the Arhinger library (Open Source BioScience) or were constructed.

Feeding RNAi in GFP::LMN-1, GFP::LMN-1 8A and LMN-1 8A CRISPR animals was performed by feeding L4 animals 12–24 hr at 20°C with RNAi bacteria as follows: mock RNAi(*ctrl*) 12–24 hr, *plk-1 (RNAi)* 10–12 hr.

### Progeny test

The percentage (%) of viability was determined by dividing the number of hatched embryos by the total number of progeny from a total of 21 wild-type and 28 *lmn-1 8A* mutant animals. Animals were maintained on NGM plates containing OP50 bacteria at 23°C.

### Generation of the *lmn-1* 8A strain

The *lmn-1* 8A strain was generated by CRISPR as described (*Paix et al., 2015*). Aminoacids 21, 22, 24, 32 (1st cluster) were mutated to alanines using the following crRNA: 5' TCGCTAAGCAACAA TGGAGGAGG 3' (the underlined sequence marks the PAM sequence). The following repair template, which includes the mutations of Ser21, 22, 24 and 32 to alanine (GCT), was used:

5′GTAGTTCTCGTATTGTTACGCTAGAGCGCTCAGCGAATGCTGCTCTAGCTAACAATGGAG-GAGGCGACGATGCGCTGTGAGTTTTTATTTTGCTTCTCTAGTTCCTGCACTCATAAACGCATTTCTTGTGTTATCTAAAAAAAAACTCGAT 3′.

Aminoacids 397, 498, 403, 405 (2nd cluster) were mutated to alanine using the following crRNA: 5′ CACGTCTCGTTTTCATCCGGAGG 3′ (the underlined sequence marks the PAM sequence). The following repair template, which includes the mutations of Thr397, Ser398, 403, and 405 to alanine (GCT), was used:

5′ AACTCAAGACCTACCAAGCTCTCCTTGAGGGTGAGGAGGAGCGTCTCAATCTTACTCAG-GAGGGGCCACAAAACGCTGCTGTTCATCACGTCGCTTTTGCTTCCGGAGGAGCAAGCGCTCAGCGCGGAGTGAAGCGTCGTCGCGTTGTCG 3′.

Note that in both cases the repair templates include also a silent restriction site for genotyping purposes (see below).

For following the successful events a PCR followed by restriction was performed for both cases.

For the 1st cluster the following primers were used:
Fwd: 5′ ATGTCATCTCGTAAAGGTACTCG 3′
Rev: 5′ ACTCACGTCGATGTAAGTGGC 3′.
The PCR product was digested by the enzyme HhaI which cut only in successful CRISPR events.
For the 2nd cluster the following primers were used:
Fwd: 5′ AACCTCGTCGATTCCAACTGG 3′
Rev: 5′ TCGACGACAAGGATGCTCG 3′
The PCR product was digested by the enzyme HaeIII which cut only in successful CRISPR events.

## Mass spectrometry

### Tryptic digestion

Proteins were digested in gel. Briefly, excised bands were distained twice for 15 min in a solution containing 50 mM NH4HCO3% and 50% acetonitrile, dehydrated in acetonitrile and dried for 10 min at 37°C. Proteins in spots were reduced with DTT 10 mM, alkyled with iodoacetamide 55 mM and then digested overnight at 37°C by sequencing grade trypsin (12.5 µg/ml, Promega) in 20 µl of NH4HCO3 25 mM and extracted from gel.

### LC-MS/MS acquisition

Digested peptides solution was desalted using ZipTip µ-C18 Pipette Tips (Millipore). Peptides were analyzed by an Orbitrap Fusion Tribrid mass spectrometer (Thermo Fisher Scientific, San Jose, CA) equipped with a Thermo Scientific EASY-Spray nanoelectrospray ion source and coupled to an Easy nano-LC Proxeon 1000 system (Thermo Fisher Scientific, San Jose, CA). Chromatographic separation of peptides was performed with the following parameters: pre-column Acclaim PepMap100 (2 cm, 75 µm i.d., 3 µm, 100 Å), column LC EASY-Spray C18 column (75 cm, 75 µm i.d., 2 µm, 100 Å), 300 nl/min flow, gradient rising from 95% solvent A (water, 0.1% formic acid) to 35% B (100% acetoni-trile, 0.1% formic acid) in 98 min. Peptides were analyzed in the orbitrap cell in full ion scan mode at a resolution of 120000 (at m/z 400) within a mass range of m/z 350–1550. Fragments were obtained with a Higher-energy Collisional Dissociation (HCD) activation with a collisional energy of 27%, and a quadrupole isolation width of 1.6 Da. MS/MS data were acquired in the linear ion trap in top-speed mode, with a dynamic exclusion of 50 s and a repeat duration of 60 s. The maximum ion accumula-tion times were set to 250 ms for MS acquisition and 60 ms for MS/MS acquisition in parallelization mode.

### LC-MS/MS data processing

For the identification step, all MS and MS/MS data were processed with the Proteome Discoverer software (Thermo Scientific, version 2.2) and with the Mascot search engine (Matrix Science, version 5.1). The mass tolerance was set to six ppm for precursor ions and 0.02 Da for fragments. The fol-lowing modifications were allowed: Acetyl (N-term), Methylation (M), Phospho (ST), Phospho (Y). The peptide identification was performed on *Caenorhabditis elegans* protein database from

Swissprot. Peptide identifications were validated using a 1% FDR (False Discovery Rate) threshold calculated with the Percolator algorithm.

## Biochemical assays

Western blot analysis was performed using standard procedures.

### Purification of GST-LMN-1 fragments

The expression of the N-terminal head [H] and C-terminal tail fragments [T] and [T1] of LMN-1 N-terminally fused to GST was induced by the addition of 1 mM of isopropyl β-D-thiogalactopyranoside (IPTG) to 1 L cultures of exponentially growing *Escherichia coli* BL21 DE3 pLysS (Invitrogen) strain (OD = 0.6), before incubation for 3 hr at 25°C. After pelleting by centrifugation, the bacteria were resuspended in lysis buffer (0.5 M NaCl, 5% glycerol, 50 mM Tris/HCl pH 8, 1X protease inhibitors), before lysis by sonication. The soluble portion of the lysate was loaded on a 1 ml GST-Trap Column (GE, healthcare). The column was washed with ten volumes of lysis buffer, and bound proteins were eluted in lysis buffer containing 20 mM Glutathione pH 8. Proteins were aliquoted and flash-frozen in liquid nitrogen and stored at −80°C.

### Purification of C.e. 6×(His)-PLK-1

To produce C.e. 6×(His)-PLK-1, insect Sf9 cells were infected with appropriate baculovirus and then lysed in lysis buffer (PBS, pH 7.2, 250 mM NaCl, 30 mM imidazole, and protease and phosphatase inhibitors [Roche]), passing the cell suspension 30 times through a 21-gauge syringe needle. The lysate was clarified by centrifugation for 10 min at 16,000 g, and the supernatant was injected on HiTrap Chelating HP column loaded with nickel sulfate (GE Healthcare). Proteins were eluted by an imidazole gradient using a fast protein LC Äkta System (GE, Healthcare). Most purified elution fractions were pooled, diluted volume to volume in the lysis buffer without imidazole and containing 50% glycerol, concentrated on a centrifugal concentrator (Vivaspin VS15RH12; Vivaproducts), flash-frozen in liquid nitrogen, and stored at −80°C.

### Purification of the GST-PLK-1 PBD WT and GST-PLK-1 PBD H538A/K540M

The human GST-PLK-1 PBD WT or phosphate pincer (GST-PLK-1 PBD H538A/K540M) mutant fusion proteins were induced by the addition of 1 mM of isopropyl β-D-thiogalactopyranoside (IPTG) to 1L cultures of exponentially growing *E. coli* BL21 DE3 pLysS (Invitrogen) strain (OD = 0.6), before incubation for 3 hr at 25°C. After pelleting by centrifugation, the bacteria were resuspended in lysis buffer (10 mM Tris pH 8, 150 mM NaCl, 1 mM EDTA, 5 mM DTT, 0.05 % NP40, 1 mM PMSF, 1X protease inhibitors), before lysis by sonication. The soluble portion of the lysate was loaded on a 1 ml GST-Trap Column (GE, healthcare). After extensive washes, the bound proteins were eluted in lysis buffer containing 20 mM Glutathione pH 8. Proteins were aliquoted and flash-frozen in liquid nitrogen and stored at −80°C.

### Purification of Aurora A

The protein kinase human Aurora A was cloned as a TEV cleavable 6xHis-GST fusion, which leaves two non-native residues (GA) at the N-terminus after TEV digestion.

Transformed bacteria were incubated in LB media at 37°C under agitation and the cultures were shifted at 18°C at OD = 0.8. Protein expression was induced by addition of IPTG (final concentration 500 μM) and the cultures were incubated overnight at 18°C under agitation.

All the subsequent steps were performed at 4°C. Bacteria were pelleted and resuspended in lysis buffer (25 mM Tris pH7.5, 300 mM NaCl, 2 mM TCEP, 2 mM PMSF) and lysed in a homogenizer (Avestin). Total extract was clarified by centrifugation (30000 g, 30 min) and the resulting soluble fraction loaded 3 times on 5 mL of packed glutathione sepharose 4B medium (GE Life Sciences). After extensive washes, 1 mg of 6xHis-TEV protease was incubated overnight onto the glutathione sepharose medium. Cleaved protein was then eluted by gravity. The salt concentration was lowered to 50 mM NaCl by dilution and the protein was loaded on a 5 mL HiTrap SP HP column (GE Life Sciences) and eluted with a salt gradient on an Akta FPLC system (GE Life Sciences). The pure fractions were collected and concentrated using an ultrafiltration centrifugal protein concentrator (MerckMillipore). The protein was finally resolved on a HiLoad 16/600 Superdex 200 pg sizing column (GE Life

Sciences) equilibrated in 25 mM Tris pH7.5, 150 mM NaCl, 1 mM TCEP. The fractions containing the protein were pooled and concentrated using an ultrafiltration centrifugal protein concentrator. The purified protein was aliquoted, flash-frozen in liquid nitrogen and stored at −80°C.

## Purification of Bora[1-224] fragment

The fragment of human Bora[1-224] was cloned as a TEV cleavable 6xHis fusions, which leaves non-native residues (GAMDPEF) at the N-terminus after TEV digestion. 6xHis-Bora[1-224] was expressed in *E. coli* BL21 DE3 pLyS RIL (Thermo Fisher Scientific). Transformed bacteria were incubated in LB media at 37°C under agitation and the cultures were shifted at 18°C at OD = 0.8. Protein expression was induced by addition of IPTG (final concentration 500 µM) and the cultures were incubated overnight at 18°C under agitation. Bacteria were pelleted and resuspended in lysis buffer (25 mM Tris pH7.5, 150 mM NaCl, 20 mM imidazole, 2 mM TCEP, 2 mM PMSF) and lysed in a homogenizer at 4° C. Total extract was clarified by centrifugation (30000 g, 30 min, 4°C) and the resulting insoluble pellet was resuspended in 25 mM Tris pH7.5, 6 M guanidine HCl, 1 mM TCEP at room temperature, sonicated and clarified by centrifugation (30000 g, 30 min). The supernatant was loaded on a 5 mL HiTrap Chelating HP column loaded with nickel sulfate (GE Healthcare). After extensive washes, 6xHis-Bora was eluted by five column volumes of elution buffer (25 mM Tris pH7.5, 6 M guanidine HCl, 500 mM imidazole, 1 mM TCEP). Fractions containing the protein were pooled and dialyzed against the dialysis buffer (25 mM Tris pH7.5, 150 mM NaCl, 1 mM TCEP) for 4 hr to remove the guanidine HCl. The precipitated 6xHis-Bora[1-224] of the dialysis bag was transferred into a 50 mL canonical tube and was centrifuged at 4000 g during 10 min. The pellet containing Bora[1-224] was resuspended overnight by agitation in 50 ml of resuspending buffer (25 mM Tris pH7.5, 150 mM NaCl, 2 mM TCEP). All the subsequent steps were performed at 4°C. The solubilized 6xHis-Bora[1-224] was clarified by centrifugation at 4000 g during 10 min. The supernatant was concentrated to 5 mL using an ultrafiltration centrifugal protein concentrator and incubated overnight with 2 mg of 6xHis-TEV protease. Cleaved Bora was then loaded on a 5 mL Ni++ HiTrap Chelating HP (GE Life Sciences) to remove 6xHis-TEV and 6xHis cleaved tag. The flow through containing Bora was concentrated using an ultrafiltration centrifugal protein concentrator. Bora[1-224] was finally resolved on a Superdex 200 10/300 GL sizing column (GE Life Sciences). The fractions containing Bora[1-224] were pooled and concentrated on a centrifugal protein concentrator (Vivaspin VS15RH12; Vivaproducts), flash-frozen in liquid nitrogen, and stored at −80°C.

## Ce-lamin purification

Ce-lamin was cloned into a modified pET28b vector. The Ce-lamin production and purification, as well as filaments assembly, was performed essentially as described (*Foeger et al., 2006*).

The 6xHis-TEV-LMN-1 was expressed by adding 1 mM of isopropyl β-D-thiogalactopyranoside (IPTG) to 1L cultures of exponentially growing *E. coli* BL21 DE3 pLysS (Invitrogen) strain (OD = 0.6), before incubation for 3 hr at 25°C. Bacteria were harvested by centrifugation at 6000 rpm for 10 min. Then the pellet was resuspended in urea lysis buffer (8 M urea, 100 mM NaCl, 10 Mm Tris–HCl, pH 8, 1 mM 2-mercaptoethanol). The bacterial suspension was sonicated four times for 30 s each at 50% pulse, and cell debris were removed by centrifugation at 11 000 rpm for 30 min at room temperature. The supernatant was loaded on a 1 ml HiTrap HP Colum (GE Healthcare). The column was washed with ten volumes of lysis buffer and the bound protein was eluted with the elution buffer (8M urea and 1M imidazole). For removal of the His-tag, lamin protein fractions were pooled and dialyzed into TEV-protease cleavage buffer (50 mM Tris–HCl pH 8.0, 150 mM NaCl, 1 mM DTT). 10 mg of 6xHis-TEV-LMN-1 fusion protein was incubated with 1 mg TEV protease overnight at 4°C. Processed protein was dialyzed again into the urea buffer described above and separated from TEV protease, which is 6xHis tagged, and uncleaved material via HiTrap HP Colum (GE Healthcare).

## LMN-1 filaments assembly-disassembly assays

Purified untagged Ce-lamin was assembled into head-to-tail polymers as described (*Foeger et al., 2006*). For each reaction Ce-lamin filaments (0.5 mg/ml) were dialyzed for 4 hr into a dimerization buffer (15 mM Tris–HCl pH 9, 1 mM DTT), followed by dialysis for 16 hr against a filaments assembly buffer (25 mM TrisHCl pH 7.4, 1 mM DTT). Then, MgCl$_2$ concentration was adjusted to 10 mM and ATP was added to 1 mM final concentration. LMN-1 filaments disassembly reactions were initiated

by the addition of purified C.e. 6×(His)-PLK-1 kinase, follow by incubation at 30°C for different time points (0, 20 and 40 min). The reaction was stopped by adding 10 mM EDTA, the samples were centrifuged at 100 000 g for 1 hr at 4°C and the supernatant and pellet fractions were collected. The reaction was stopped by adding 10 mM EDTA. The samples were either prepared for electron microscopy (see below) or were centrifuged at 100 000 g for 1 hr at 4°C for sedimentation analysis.

## Kinase assays

In vitro kinase assays were performed as described (*Tavernier et al., 2015*). Briefly, an equal amount of Wild-type and mutant GST-LMN-1 fragments were phosphorylated by 375 ng of C.e. 6×(His)–PLK-1 or 20 units of Cyclin B-Cdk1 (P6020L from NEB) in 30 µl total volume of the kinase buffer (1X protease inhibitors (complete cocktail from Roche), 1X phosphatase inhibitors (PhosSTOP cocktail from Roche), 2 mM ATP, 50 mM HEPES pH 7.6, 10 mM MgCl$_2$) for 40 min at 30°C under 300 rpm agitation (*Figures 1C* and *2B–2D*, *Figure 2—figure supplement 1*). For the radioactive in vitro kinase assay (*Figure 1F and G*), we used a mix of 1.7 mM of cold ATP and 5 µCi of γ-[$^{32}$P]ATP (PerkinElmer). The kinase reaction was stopped by the addition of 15 µl of Laemmli 3X, the samples were boiled and loaded on 12% SDS-PAGE. The gel was dried and exposed for autoradiography (Typhoon, GE Healthcare).

## Far-Western ligand-binding assay

GST-LMN-1 fragments phosphorylated in vitro by PLK-1 (*Martino et al., 2017*) or CyclinB-Cdk1 (*Tavernier et al., 2015*) were separated on SDS-PAGE 12% gel. The gel was then transferred to a PVDF membrane 0.45 µm during 1h30 at 90V. After saturation overnight at 4°C in blocking solution (4% milk in TBS-Tween 0.1%), the membranes were incubated with 2 µg of GST-PBD WT or the GST-PBD H538A/K540M mutant of Plk1 (version of the PBD unable to bind phosphopeptides, negative control) during 6 hr at 4°C. After extensive washing steps (every 15 min for 5 hr) with the blocking solution at 4°C, the membrane was incubated overnight at 4°C with human Plk1 antibody (1/1000) in a typical Western blot experiments to reveal the GST-PBD immobilized on the membrane.

For Phos-Tag gels, GST-LMN-1 fragments previous phosphorylated in vitro by PLK-1 were separated in a Phos-Tag100 µM SDS-PAGE 8% (*Figure 1C*) or 100 µM Phos-Tag10% SDS-PAGE (*Figure 1—figure supplement 1B*). After washing three times the Phos-tag gel with 10 mM EDTA in transfer buffer (48 mM Tris-base, 39 mM Glycine, 20% ethanol, 0.03% SDS), each gel was transferred to a PVDF membrane 0.45 µm during 1h30 at 90V. The membrane was blocked with the blocking solution (5% milk, 1% BSA in TBS-Tween 0.1%), the membranes were incubated with GST antibody (1:1000). Proteins were detected by immunoblotting and visualized by treating the blots with ECL (Millipore).

## Immunofluorescence and microscopy

Fixation and indirect immunofluorescence of *C. elegans* embryos was performed essentially as described on subbing solution-coated slides (*Joly et al., 2020*). After Freeze-crack and fixation with cold dehydrated methanol, slides were washed 3 × 5 min, blocked for 1 hr in PBS 3% BSA and incubated overnight at 4°C with primary antibodies diluted in PBS 3% BSA. Working dilutions for the primary antibodies were 1/1000 for rabbit LMN-1 antibodies. Slides were later incubated for 30 min at room temperature with secondary antibodies (anti-Rabbit) coupled to the Alexa 568 fluorophore. Next, embryos were mounted in Vectashield Mounting Medium with DAPI (Vector). Fixed embryos were imaged using a spinning disk confocal microscope with 63 × N/A 1.4 objectives. Captured images were processed using ImageJ and Adobe Photoshop.

For the analysis of the paired nuclei phenotype in live specimens by differential interference contrast (DIC) microscopy, embryos were obtained by cutting open gravid hermaphrodites using two 21-gauge needles. Embryos were handled individually and mounted on a coverslip in 3 µl of M9 buffer. The coverslip was placed on a 3% agarose pad. DIC images were acquired by an Axiocam Hamamatsu ICc one camera (Hamamatsu Photonics, Bridgewater, NJ) mounted on a Zeiss AxioImager A1 microscope equipped with a Plan Neofluar 100×/1.3 NA objective (Carl Zeiss AG, Jena, Germany), and the acquisition system was controlled by Axiovision software (Carl Zeiss AG, Jena, Germany). Images were acquired at 10 s intervals.

Live imaging was performed at 23°C using a spinning disc confocal head (CSU-X1; Yokogawa Corporation of America) mounted on an Axio Observer.Z1 inverted microscope (Zeiss) equipped with 491- and 561 nm lasers (OXXIUS 488 nm 150 mW, OXXIUS Laser 561 nm 150 mW) and sCMOS PRIME 95 camera (Photometrics). Acquisition parameters were controlled by MetaMorph software (Molecular Devices). In all cases a 63×, Plan-Apochromat 63×/1.4 Oil (Zeiss) lens was used, and approximately four z-sections were collected at 1 µm and 10 s intervals. Captured images were processed using ImageJ and Photoshop.

### Photoconversion experiment

Photoconversion of wildtype or *lmn-1 8A* embryos expressing the photoconvertible form of histone (Dendra2::H2B) in the germline was performed as described (*Bolková and Lanctôt, 2016*). By using a 'point activation' (using a ROI to select the region of interest), through illuminating the chromatin of the maternal pronucleus with the 405 nm diode laser set at power of 4 mW (4% of maximum power) and a pixel dwell time of 40µs for five repeats. A green-to-red photoconversion of the labeled chromatin was observed and followed the fate of the chromosomes during mitosis until the two-cell stage. Image stacks were processed and merged using ImageJ.

### Negative staining electron microscopy

Purified untagged Ce-lamin (0.5 mg/ml) was assembled into head-to-tail polymers as described above, LMN-1 filaments disassembly reaction was initiated by the addition of purified C.e. 6×(His)-PLK-1 kinase, follow by incubation at 30°C for 40 min in the presence of Mg/ATP. Then 3 µl of the LMN-1 assembled into head-to-tail polymers or LMN-1 filaments disassembly reaction were deposited on glow-discharged TEM grids and allowed to sediment for two mins. Excess liquid was blotted out and grids were stained in 1% uranyl acetate for 30 s, blotted and dried before observation in a Tecnai 12 (Thermofischer scientific, Eidenhoven, The Netherlands) transmission microscope at 120kV. Pictures were acquired on a Gatan Oneview camera (Roper Scientific, USA) at 0.3 to 1 nm pixel size.

### Quantification and statistical analysis

GFP::LMN-1 intensity over time was measured using the Image J software in control and RNAi conditions after background subtraction. Anaphase onset was defined as time 0. Data points on the graphs are the mean of the normalized GFP intensity measurements in control and RNAi conditions for the same define region of interest (ROI). To allow direct comparison between control and RNAi conditions, the average signal intensity of GFP::LMN-1 at the NE 120 s before anaphase was arbitrarily defined as 1. The results are presented as means ± SEM. In all graphs, data were compared by Mann-Whitney test. All calculations were performed using GraphPad Prism version 6.00 for Mac OS X, GraphPad Software, La Jolla California USA, www.graphpad.com.

## Acknowledgements

We thank P Moussounda and R Servouze for help with media preparation. We thank V Archambault, B Cisneros, B Lanctot for strains and reagents. We thank X Baudin and V Contremoulins for microscopy data acquisition and analysis. We are grateful to Angela Graf for technical assistance. We thank R Karess and V Doye for fruitful discussions during the course of the study and R Karess for critical reading of the manuscript. We acknowledge the ImagoSeine core facility of the Institut Jacques Monod, member of IBiSA and France-BioImaging (ANR-10-INBS-04) infrastructures and the Institut Jacques Monod 'Structural and Functional proteomic platform'. Some strains were provided by the Caenorhabditis Genetics Center (CGC), which is funded by NIH Office of Research Infrastructure Programs (P40 OD010440). GVA was supported by the Labex 'Who am I?' Laboratory of Excellence No. ANR-11-LABX-0071, the French Government through its Investments for the Future program operated by the French National Research Agency (ANR) under Grant no. ANR-18-IDEX-0001 and by the CONACYT grant CVU 364106. SNN is supported by a PhD fellowship from the Ministry of Research. NJ is supported by a funding 'Dynamic Research' from ANR-18-IDEX-0001, IdEx Université de Paris. Work in the laboratory of LP is supported by grants from 'Agence Nationale pour la Recherche' (ANR, France - ANR-17-CE13-0011) and by the 'Ligue Nationale Contre le Cancer'

(Equipe labéllisée, France). Verena Jantsch acknowledges funds from the Austrian Science Fund (FWF project no. P 32903-B).

## Additional information

### Funding

| Funder | Grant reference number | Author |
| --- | --- | --- |
| Agence Nationale de la Recherche | ANR-17-CE13-0011 | Lionel Pintard |
| Ligue Contre le Cancer | Labellisation | Lionel Pintard |
| Consejo Nacional de Ciencia y Tecnología | CVU364106 | Griselda VELEZ-AGUILERA |
| Austrian Science Fund | P32903-B | Verena Jantsch |
| Agence Nationale de la Recherche | ANR-11-LABX-0071 | Griselda VELEZ-AGUILERA |

The funders had no role in study design, data collection and interpretation, or the decision to submit the work for publication.

### Author contributions

Griselda Velez-Aguilera, Conceptualization, Data curation, Formal analysis, Validation, Investigation, Methodology, Writing - review and editing; Sylvia Nkombo Nkoula, Batool Ossareh-Nazari, Jana Link, Nicolas Joly, Formal analysis, Investigation, Methodology; Dimitra Paouneskou, Data curation, Investigation, Methodology; Lucie Van Hove, Methodology; Nicolas Tavernier, Resources, Methodology; Jean-Marc Verbavatz, Investigation, Methodology; Verena Jantsch, Conceptualization, Formal analysis, Supervision, Funding acquisition, Validation, Project administration, Writing - review and editing; Lionel Pintard, Conceptualization, Resources, Data curation, Formal analysis, Supervision, Funding acquisition, Validation, Investigation, Visualization, Methodology, Writing - original draft, Project administration, Writing - review and editing

### Author ORCIDs

Griselda  Velez-Aguilera (ID) https://orcid.org/0000-0002-9662-8833
Verena Jantsch (ID) http://orcid.org/0000-0002-1978-682X
Lionel Pintard (ID) https://orcid.org/0000-0003-0286-4630

### Decision letter and Author response

Decision letter https://doi.org/10.7554/eLife.59510.sa1
Author response https://doi.org/10.7554/eLife.59510.sa2

## Additional files

### Supplementary files

- Supplementary file 1. LC-MS/MS analysis of phosphorylated LMN-1 in vitro.

- Supplementary file 2. Sequence analysis of LMN-1 phospho-sites. Phosphorylated serine and threonine residues (pS and pT) are highlighted in yellow and green respectively. Most phosphosites match the previously described Plk1 consensus motifs [L($\Phi$)-D/E/N/Q-X-pS/pT-L($\Phi$) or p[S/T]-F] (*Santamaria et al., 2011*; *Kettenbach et al., 2011*). D/E/N/Q residues are highlighted in blue, hydrophobic ($\Phi$) in orange and F. in red.

- Supplementary file 3. Accession numbers of lamin protein sequences.

- Supplementary file 4. Oligonucleotides used in this study.

- Transparent reporting form

## Data availability

All data generated or analysed during this study are included in the manuscript and supporting files. Source data files have been provided.

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
