## [Decision Letter]

**Acceptance summary:**

Your study identifies a new role for the Plk1 protein kinase in disassembly of the nuclear lamina, adding to our understanding of how the cell architecture is reshaped in mitosis.

**Decision letter after peer review:**

Thank you for submitting your article "PLK-1 promotes the merger of the parental genome into a single nucleus by triggering lamina disassembly" for consideration by *eLife*. Your article has been reviewed by three peer reviewers, and the evaluation has been overseen by a Reviewing Editor and Anna Akhmanova as the Senior Editor. The following individuals involved in review of your submission have agreed to reveal their identity: Brian Burke (Reviewer #3).

The reviewers have discussed the reviews with one another and the Reviewing Editor has drafted this decision to help you prepare a revised submission.

Summary:

In the manuscript “PLK-1 promotes the merger of the parental genome into a single nucleus by triggering lamina disassembly”, the authors investigate the role of the protein kinase PLK-1 in the fusion of the maternal and paternal pronuclei in *C.elegans* zygotes. It was previously known that pronuclear fusion depends on PLK-1 activity and that LMN-1 is a PLK-1 target in vivo. The authors now demonstrate that PLK-1 can phosphorylate recombinant fragments of *C. elegans* LMN-1 in vitro, and identify putative PLK-1 target sites by mass spectrometry. Using a phosphorylation-deficient mutant of LMN-1, the authors demonstrate that LMN-1 phosphorylation drives the timely disassembly of the nuclear lamina in the zygote as a prerequisite for the mixing of the parental genomes. The manuscript presents a set of well-controlled experiments that are of commendable quality and in support of the proposed model.

All the reviewers asked that you clarify the specificity of the Plk1 kinase assays. They also agreed that the manuscript would benefit from rewriting to make it clearer focusing on Plk1's role in lamina disassembly and make the parts on "merger of parental genomes" more compact and concise since such binucleate phenotypes have already been characterised extensively. Moreover, mentioning the role of microtubules, but then not going into the details, e.g. by visualizing microtubules themselves, is more confusing than helpful – these data may have a better use as a basis for a new study.

It would be helpful to provide a clearer description of processes in the early embryo in *C. elegans*. There are sections of the Introduction that are confusing unless you are a *C. elegans* specialist. In particular, gap formation in the opposed nuclear membranes depends on the alignment of chromosomes at the metaphase plate and for this to happen limited NEBD must occur in regions of the NEs closest to the spindle poles, which raises the question of how NEBD is spatially regulated. In the absence of PLK-1 the membrane gap does not appear, resulting in the formation of paired maternal and paternal nuclei in the two cell embryo but the chromosomes are still segregated, indicating that limited NEBD must still occur close to the spindle poles. Clearly, mechanisms of NEBD are spatially segregated. Even in wild type embryos, the distal margins of the two NEs survive until the metaphase-anaphase transition. Is there a mechanical aspect to this? Similarly, for the apparent flattening of the two opposed NEs: is there any evidence of physical interactions between the two outer nuclear membranes, perhaps analogous to that which occurs between Golgi cisternae? Another possibility would be interaction between outer nuclear membrane KASK-domain proteins. Microtubulule-dependent associations between opposed UNC-83 molecules for instance. Do we know if SUN or KASH domain proteins are concentrated in this contact region? It goes without saying that there have to be interactions between the two ONMs if the fusion model described in the Rahman et al., 2020, paper proves to be correct.

A few more comments on LMN-1 might also be useful. Is it more closely related to vertebrate B-type lamins, is it farnesylated, does it remain membrane-associated during mitosis, etc?

Revisions expected in follow-up work:

(1) in vitro kinase assays may suffer from a lack of specificity. In light of this fact, it must be noted that phosphorylation by PLK-1 in the presented in vitro assays is only suppressed if all available Ser and Thr residues are mutated to Ala. And, as mentioned by the authors, some of the identified phosphosites do not follow the consensus sequence of PLK-1 phosphorylation, especially not the broadened consensus defined by Santamaria et al., 2010 (L(Φ)(E/N/D(Q))X(S/T)L(Φ)) or the previously accepted consensus (D/E)X(S/T)Φ (Nakajima et al., 2003). The consensus sequence of PLK-1 should be shown as part of Figure 1. The comparison shown in Supplementary file 2 may overestimate the fit to the consensus by using the most minimal motifs. In the table, it should also be indicated which sites were identified as PLK-1 target sites in vivo (Link et al., 2018). How specific is Plk1 as compared to cdk1 or Aurora, for example, in phosphorylating these sites? How strongly are these sites phosphorylated by Plk1 as compared to phosphorylation of cdk1-sites by cdk1?

(2) In the text it is stated that depletion of PLK-1 in the GFP::LMN-1 8A mutant strain exacerbates the lamina disassembly defect (Figure 3). This should not be the case if the authors had identified all relevant PLK-1 phosphorylation sites. The authors ascribe this additional effect on the lamina to the remaining seven phosphosites that were phosphorylated in their in vitro assay, but not mutated in this experiment. Since these seven additional phosphosites have not been shown to be phosphorylated by PLK-1 in previous studies in vivo studies, one might wonder whether the exacerbation of the lamina disassembly effect can be ascribed to something other than these 15 putative phosphosites. In order to exclude this possibility, it would have been helpful to analyze a 15A mutant. In any case, the effect of PLK-1 depletion in the mutant worm is most striking after anaphase onset. The authors should discuss this observation.

3) Figure 1 shows that Plk1 can efficiently phosphorylate lamin on several sites. However, the figure lacks data that would allow a relative comparison of how efficient and specific this phosphorylation really is.

[Editors' note: further revisions were suggested prior to acceptance, as described below.]

Thank you for submitting your article “PLK-1 promotes the merger of the parental genome into a single nucleus by triggering lamina disassembly" for consideration by *eLife*. Your article has been reviewed by two peer reviewers, and the evaluation has been overseen by a Reviewing Editor and Anna Akhmanova as the Senior Editor. The reviewers have opted to remain anonymous.

The reviewers have discussed the reviews with one another and the Reviewing Editor has drafted this decision to help you prepare a revised submission.

The reviewers support the publication of your paper in principle but they, and the editor, are disappointed that you made little effort to address the specificity of the in vitro kinase assays that was a major concern for all the reviewers. Please address this point with new data in your revision. You should also provide a control, treated compared to untreated, for Figure 5C.

---

## [Author Response]

Revisions for this paper:All the reviewers asked that you clarify the specificity of the Plk1 kinase assays.

We thank the reviewers for bringing up this important point. We fully understand the concern that the in vitro kinase assays may suffer from a lack of specificity, but available data in the literature indicate that Plk1 generally phosphorylates its substrates on the same sites in vitro and in vivo. In fact, Plk1 consensus sites were originally defined using in vitro kinase assays with purified Plk1 and GST-Cdc25C peptides (Nakajima et al., 2003), exactly as we did here with GST-LMN-1 fragments. Later Plk1 consensus sites were refined by combining phosphoproteomic studies in human cells with mass-spectrometry-based in vitro kinase assays using naturally occurring peptides (Santamaria et al., 2010, Kettenbach et al., 2011). Overall, these studies have extensively used in vitro kinase assays to validate Plk1 sites and to define the consensus sequence.

Furthermore, it should be noted that PLK-1 does not phosphorylate GST which itself contains multiple serine and threonine residues. Likewise, the GST-LMN-1[H] 9A fragment, which contains four serine and two threonine residues at the beginning of the LMN-1 sequence is not phosphorylated by PLK-1, indicating the specificity of the assay. We also used *C. elegans* PLK-1 kinase in our assay to further ensure specificity. We also clearly demonstrate that some of these phosphosites are self-priming sites. Finally, we clearly show that S32, which is phosphorylated in vitro by PLK-1 is also modified in vivo in a PLK-1-dependent manner. For all these reasons we are confident that our assay is relatively specific.

Regarding the Plk1 consensus site, we note that it has been defined using human proteins and we currently do not know whether it is exactly the same consensus with *C. elegans* proteins. In addition, as emphasized by Santamaria et al., even the broadened consensus L(F)(E/N/D(Q))X(S/T)L(F) does not account for all observed Plk1 substrates. Kettenbach et al., 2011 have identified variations of this motif (Kettenbach et al. Science Signaling 2011, Figure S11B).

They also agreed that the manuscript would benefit from rewriting to make it clearer focusing on Plk1's role in lamina disassembly and make the parts on "merger of parental genomes" more compact and concise since such binucleate phenotypes have already been characterised extensively.

The main focus of our study was to decipher how PLK-1 promotes the merger of the parental chromosomes by identifying its key target(s) at the nuclear envelope.

As PLK-1 is expected to phosphorylate numerous substrates during mitosis, the paired nuclei phenotype could have resulted from the non-phosphorylation of multiple PLK-1 targets. Here we not only identify the lamina (LMN-1) as a PLK-1 target but we demonstrate that the expression of LMN-1 with point mutations on PLK-1 sites is sufficient to recapitulate the paired nuclei phenotype resulting from *plk-1* inactivation. We thus demonstrate that depolymerization of the lamina is a key and essential step towards the merging of the parental genomes.

While it is true that the paired nuclei phenotype has been already described, only one study demonstrated that parental chromosomes remain physically separated in embryo presenting this phenotype (Rahman et al., 2015). Thus to prove our point it was critical to demonstrate unequivocally that the expression LMN-1 with point mutations on PLK-1 sites prevents the merging of the parental genomes and thus that LMN-1 is a key target.

We discuss extensively the implication of the identification of LMN-1 as a new Plk1 target in *C. elegans* and possibly in other systems in the Discussion section. For all these reasons, we would rather prefer to keep the organization of the manuscript as it is. Nevertheless, we have slightly modified the Abstract to better balance the two main findings of the study.

Moreover, mentioning the role of microtubules, but then not going into the details, e.g. by visualizing microtubules themselves, is more confusing than helpful – these data may have a better use as a basis for a new study.

We fully agree with this comment. Consequently, we have decided to remove the last section on microtubules. This part does not change the main message of our study. We will publish it separately.

It would be helpful to provide a clearer description of processes in the early embryo in *C. elegans*. There are sections of the Introduction that are confusing unless you are a *C. elegans* specialist. In particular, gap formation in the opposed nuclear membranes depends on the alignment of chromosomes at the metaphase plate and for this to happen limited NEBD must occur in regions of the NEs closest to the spindle poles, which raises the question of how NEBD is spatially regulated. In the absence of PLK-1 the membrane gap does not appear, resulting in the formation of paired maternal and paternal nuclei in the two cell embryo but the chromosomes are still segregated, indicating that limited NEBD must still occur close to the spindle poles. Clearly, mechanisms of NEBD are spatially segregated. Even in wild type embryos, the distal margins of the two NEs survive until the metaphase-anaphase transition. Is there a mechanical aspect to this? Similarly, for the apparent flattening of the two opposed NEs: is there any evidence of physical interactions between the two outer nuclear membranes, perhaps analogous to that which occurs between Golgi cisternae? Another possibility would be interaction between outer nuclear membrane KASK-domain proteins. Microtubulule-dependent associations between opposed UNC-83 molecules for instance. Do we know if SUN or KASH domain proteins are concentrated in this contact region? It goes without saying that there have to be interactions between the two ONMs if the fusion model described in the Rahman et al., 2020, paper proves to be correct.

We apologize for not having been sufficiently clear on the description of NEBD in the one cell embryo. We have modified the Introduction accordingly. Clearly mechanisms of NEBD are spatially regulated in the early embryo and the localization of PLK-1 is probably one of the mechanisms dictating this timing.

The mechanisms allowing the flattening of the two opposed NEs are unknown and whether the two membranes physically interact before fusion and how they do it is not understood. SUN and KASH domain proteins are located at the NE but are not particularly enriched in this contact region.

A few more comments on LMN-1 might also be useful. Is it more closely related to vertebrate B-type lamins, is it farnesylated, does it remain membrane-associated during mitosis, etc?

It appears that we have not been sufficiently clear on this point either. We have included additional comments to specify that LMN-1 is the unique B-type lamin ortholog in *C. elegans*.

Revisions expected in follow-up work:1) in vitro kinase assays may suffer from a lack of specificity. In light of this fact, it must be noted that phosphorylation by PLK-1 in the presented in vitro assays is only suppressed if all available Ser and Thr residues are mutated to Ala.

As mentioned above, the GST-LMN-1[H] 9A fragment, which still contains four serine and two threonine residues at the beginning of LMN-1 sequence is not phosphorylated by PLK1. GST, used as a control in these experiments, is also not phosphorylated by Plk1.

And, as mentioned by the authors, some of the identified phosphosites do not follow the consensus sequence of PLK-1 phosphorylation, especially not the broadened consensus defined by Santamaria et al., 2010 (L(Φ)(E/N/D(Q))X(S/T)L(Φ)) or the previously accepted consensus (D/E)X(S/T)Φ (Nakajima et al., 2003).

That is true: some of the identified phosphosites do not follow the consensus defined by Santamaria et al. but it is important to mention that Santamaria et al. emphasized in their study that even the broadened consensus L(F)(E/N/D(Q))X(S/T)L(F) does not account for all observed Plk1 substrates. Kettenbach et al., 2011 identified additional consensus sites including pS/pT-F or pS/pT with a hydrophobic residue in position +1 (Kettenbach et al., 2011, Figure S11B). There are several variations of these consensus sequences, as noted in these studies.

The consensus sequence of PLK-1 should be shown as part of Figure 1.

We have now included the consensus sequence of PLK-1 in the main text but not in Figure 1. Just adding the consensus sequence in the Figure will not help the reader unless we insert the full Supplementary file 2 that we have extensively revised (see next comment).

The comparison shown in Supplementary file 2 may overestimate the fit to the consensus by using the most minimal motifs.

We have revised the Supplementary file 2 to compare the surrounding sequence of each phosphosites to the broaden consensus sites defined by Santamaria et al. or Kettenbach et al. It should be noted that several contiguous phosphorylated residues such as S21 and S22 are embedded in a consensus site.

In the table, it should also be indicated which sites were identified as PLK-1 target sites in vivo (Link et al., 2018).

Please note that Link et al. did not show that these sites are PLK-1 targets in vivo but CHK2 and PLK-2 targets in the *C. elegans* germline. We show here that LMN-1 S32 is a PLK-1 target in the early embryo. As suggested, we now indicate in the Supplementary file 2 the sites that have been found to be phosphorylated in vivo.

How specific is Plk1 as compared to cdk1 or Aurora, for example, in phosphorylating these sites? How strongly are these sites phosphorylated by Plk1 as compared to phosphorylation of cdk1-sites by cdk1?

Extensive phosphoproteomic studies have shown that these three kinases clearly phosphorylate different motifs. As mentioned in the text, LMN-1 does not contain a single S/T-P site typically phosphorylated by Cyclin-dependent kinases nor R/KpS/pT or R/K-XpS/pT motifs phosphorylated by Aurora kinases. Whether Plk1 and Cdk1 phosphorylates their sites with equal efficiency is, to the best of our knowledge, not known but the fact that some phosphosites in the head domain of LMN-1 are also self-priming sites clearly increases the efficiency of phosphorylation by concentrating the Plk1 kinase in the vicinity of its substrate. Accordingly, our data suggest that the LMN-1 head domain, which contains two self-priming sites, is more efficiently phosphorylated than the LMN-1 tail domain which lacks such sites.

2) In the text it is stated that depletion of PLK-1 in the GFP::LMN-1 8A mutant strain exacerbates the lamina disassembly defect (Figure 3). This should not be the case if the authors had identified all relevant PLK-1 phosphorylation sites. The authors ascribe this additional effect on the lamina to the remaining seven phosphosites that were phosphorylated in their in vitro assay, but not mutated in this experiment. Since these seven additional phosphosites have not been shown to be phosphorylated by PLK-1 in previous studies in vivo studies, one might wonder whether the exacerbation of the lamina disassembly effect can be ascribed to something other than these 15 putative phosphosites. In order to exclude this possibility, it would have been helpful to analyze a 15A mutant. In any case, the effect of PLK-1 depletion in the mutant worm is most striking after anaphase onset. The authors should discuss this observation.

This is indeed a very important point. We are in the process of generating *C. elegans* lines to address this important issue. We now emphasize this point in the Discussion as suggested.

3) Figure 1 shows that Plk1 can efficiently phosphorylate lamin on several sites. However, the figure lacks data that would allow a relative comparison of how efficient and specific this phosphorylation really is.

As discussed above Figure 1 shows that the LMN-1 head is more efficiently phosphorylated than the LMN-1 tail domain.

[Editors' note: further revisions were suggested prior to acceptance, as described below.]

The reviewers support the publication of your paper in principle but they, and the editor, are disappointed that you made little effort to address the specificity of the in vitro kinase assays that was a major concern for all the reviewers. Please address this point with new data in your revision. You should also provide a control – treated compared to untreated – for Figure 5C.

We understand the concerns of the reviewers that our in vitro kinase assay may suffer from a lack of specificity but we believe that we have included the appropriate controls in these in vitro experiments. To summarize, we show that PLK-1 phosphorylates GST-LMN-1 fragments but not GST (control) on multiple PLK-1 consensus sites and that mutation of these sites eliminates phosphorylation by PLK-1 in vitro. In addition, we show that LMN-1 contains functional polo-docking sites, which is the signature of PLK-1 substrates.

To try to address the concerns of the reviewers, we have tested whether other mitotic kinases, including Aurora A or Cyclin B-Cdk1, can phosphorylate the LMN-1 head domain. As a control, we have used a fragment of Bora (Bora^1-224^), which is known to be phosphorylated by Cyclin B-Cdk1 and Aurora A kinases. As shown in the revised Figure 1 supplement 1B, PLK-1, but not Cyclin B-Cdk1 or Aurora A, induced a mobility shift of GST-LMN-1 head on Phos-tag gels. As a positive control, we show that both Cyclin B-Cdk1 and Aurora A induced a mobility shift of Bora^1-224^ in these conditions. These data indicate that PLK-1, but not Aurora A or Cyclin B-Cdk1, phosphorylates the LMN-1 head. We hope that these new experiments address the concerns of the reviewers. We cite these results as follows:

“In a similar assay, the other mitotic kinases Aurora A and Cyclin B-Cdk1 failed to induce a mobility shift of GST-LMN-1 [H] (Figure 1—figure supplement 1B) indicating that PLK-1, but not these other mitotic kinases, phosphorylates LMN-1 in vitro, possibly at multiple residues.”

“….lamins in nematodes, do not harbor any Cdk S/T-P motifs in the head or in the tail domains (Figure 2A). Proline residues, highlighted in red in the protein sequence alignment presented in Figure 2A, are strikingly absent in the head and tail domains of lamins from nematodes, consistent with the observation that Cyclin B-Cdk1 failed to phosphorylate GST-LMN-1[H] (Figure 1—figure supplement 1B).”

It was also requested that we perform EM analysis of Lamin filaments after treatment with PLK-1. We now provide in Figure 5—figure supplement 1 additional EM images of LMN-1 filaments untreated (+ Mg/ATP, left panels) versus treated with PLK-1 (+ Mg/ATP PLK-1, right panels). We present images taken at different magnifications. Whereas a dense network of LMN-1 filaments was visible in the untreated condition, the network was no longer detectable in the treated sample, even though some small LMN-1 filaments were visible, which agrees with the biochemical data presented Figure 5D, indicating that part of LMN-1 remains insoluble after treatment with the PLK-1 kinase.

We cite these observations as follows:

“To corroborate these observations, we used electron microscopy to examine LMN-1 filaments 40 min after incubation with PLK-1. In the absence of PLK-1, a dense meshwork of LMN-1 filaments was detectable but this network was no longer visible in the presence of PLK-1 (Figure 5—figure supplement 1). Taken together, these results indicate that LMN-1 phosphorylation by PLK-1 promotes lamina disassembly in vitro.”